# Selective photoelectrochemical oxidation of glucose to glucaric acid by single atom Pt decorated defective TiO$_2$

Zhangliu Tian [1,7] ✉, Yumin Da[1,2,7], Meng Wang[1,2], Xinyu Dou[1], Xinhang Cui[1], Jie Chen[2,3], Rui Jiang[4], Shibo Xi[5], Baihua Cui[1,2], Yani Luo[1,2], Haotian Yang[1,2], Yu Long[1,2], Yukun Xiao[1,2] & Wei Chen [1,2,3,6] ✉

Photoelectrochemical reaction is emerging as a powerful approach for biomass conversion. However, it has been rarely explored for glucose conversion into value-added chemicals. Here we develop a photoelectrochemical approach for selective oxidation of glucose to high value-added glucaric acid by using single-atom Pt anchored on defective TiO$_2$ nanorod arrays as photoanode. The defective structure induced by the oxygen vacancies can modulate the charge carrier dynamics and band structure, simultaneously. With optimized oxygen vacancies, the defective TiO$_2$ photoanode shows greatly improved charge separation and significantly enhanced selectivity and yield of C$_6$ products. By decorating single-atom Pt on the defective TiO$_2$ photoanode, selective oxidation of glucose to glucaric acid can be achieved. In this work, defective TiO$_2$ with single-atom Pt achieves a photocurrent density of 1.91 mA cm$^{-2}$ for glucose oxidation at 0.6 V versus reversible hydrogen electrode, leading to an 84.3 % yield of glucaric acid under simulated sunlight irradiation.

Biomass conversion into building-block chemicals coupled with clean fuel generation using renewable energy is a promising strategy to alleviate the dependence on fossil fuels[1]. As one of the most important platform molecules of biomass, glucose has triggered growing attention due to its various value-added products such as glucaric acid (GLA), gluconic acid (GLU), sorbitol, and 5-hydroxymethylfurfural[2]. GLA as a building block chemical in a bio-based economy has been identified as one of the 12 top value-added chemicals by the US Department of Energy in 2004[3]. GLA and its derivatives are widely used for the production of a variety of commodity products for healthcare including cholesterol reduction and cancer chemotherapy[4]. The high functionalization of GLA leads to high economic value and huge

market demand, which is predicted to be USD 1.30 billion by 2025 million according to the report of Grand View Research, Inc[2]. However, the productions of GLA by current methods are insufficient to support the large market demand. Currently, the productions of GLA are obtained either by microbial fermentation or chemical oxidation[5,6]. The chemical oxidation is usually realized under harsh conditions, such as with HNO$_3$ or oxidant, at high temperature, or under high pressure. Various by-products are formed during the chemical oxidation, and expensive separation for the extraction of GLA is required. Besides, selective catalytic conversion of glucose to GLA has also been achieved by electrochemical or thermal oxidation on noble metal catalysts (e.g., Au, Ag, Pd, and Pt) with additional

[1]Department of Chemistry, National University of Singapore, 3 Science Drive 3, Singapore 117543, Singapore. [2]Joint School of National University of Singapore and Tianjin University, International Campus of Tianjin University, Binhai New City, Fuzhou 350207, China. [3]Department of Physics, National University of Singapore, 2 Science Drive 3, Singapore 117542, Singapore. [4]School of Materials Science and Engineering, Tianjin University, Tianjin 300072, China. [5]Institute of Sustainability for Chemicals, Energy and Environment, Agency for Science, Technology and Research (A*STAR), 1 Pesek Road, Jurong Island, Singapore 627833, Singapore. [6]Centre for Hydrogen Innovations, National University of Singapore (Singapore), E8, 1 Engineering Drive 3, Singapore 117580, Singapore. [7]These authors contributed equally: Zhangliu Tian, Yumin Da. ✉e-mail: tianzl@nus.edu.sg; phycw@nus.edu.sg

oxidants, which are still cost-intensive and with limited efficiency for GLA production[7].

Photoelectrochemical (PEC) oxidation provides another promising strategy for GLA production, which can be operated under mild conditions by eliminating the use of hazardous chemical oxidants or high-pressure $O_2$[8]. Besides, PEC cell is the way to drive reactions through the photo-induced electrons and holes directly from the sunlight with a very low applied potential, which shows good potential for energy saving and environmental protection[9–11]. Therefore, PEC GLA production is proposed to be a green strategy to produce value-added chemicals with clean fuel by combining the green energy source with renewable feedstock[12]. Although the noble metal-based catalysts have shown high selectivity towards GLA production during the anodic oxidation of glucose, the high cost restrains the actual application[3,13–15]. Single-atom catalysts with well-defined active centers and maximum atom utilization can greatly reduce the amount of noble metals, thus significantly reducing the cost[16,17]. Therefore, single-atom noble metals modified photoanodes are expected to achieve PEC glucose oxidation with high selectivity and yield.

In this work, selective oxidation of glucose to GLA is realized on the photoanode of defective $TiO_2$ decorated with single-atom Pt via a PEC strategy (Fig. 1). The defective $TiO_2$ is fabricated by electrochemical reduction of $TiO_2$ nanorod arrays, and further deposition of single-atom Pt is achieved by atomic layer deposition. The disordered structure in the defective $TiO_2$ has been verified to promote charge separation and transportation, along with the modulation of the oxidation products (Fig. 1). The single-atom Pt on defective $TiO_2$ has been evidenced to regulate the selectivity of the GLA by accelerating the GLU oxidation. With the as-prepared photoanodes, a high yield of 84.3% for GLA is achieved at 0.6 $V_{RHE}$ under simulated sunlight illumination.

## Results and discussion

### Synthesis and characterizations of nanorod arrays with Pt atoms

$TiO_2$ nanorod arrays (NRAs) were directly grown on fluorine-doped tin oxide (FTO) substrates via a typical hydrothermal method followed by annealing in air (Fig. 2a)[18]. Scanning electron microscopy (SEM) was first employed to characterize the morphology and structure of the sample. As illustrated in Fig. 2b, c, the obtained film on the FTO substrate is composed of NRAs, which are vertically and evenly distributed on the FTO substrate. The length of the nanorod is about 1.3 μm. The statistically determined diameter distribution (Inset of Fig. 2c) shows that the average diameter of the nanorod is 120 nm with peaks in the range of 90–150 nm. The x-ray diffraction (XRD) pattern indicates that all the diffraction peaks of (002), (211), and (101) from the NRAs are following the rutile phase (JCPDS No. 21-1276), confirming the formation of NRAs in the rutile $TiO_2$ crystal structure (Fig. 2d).

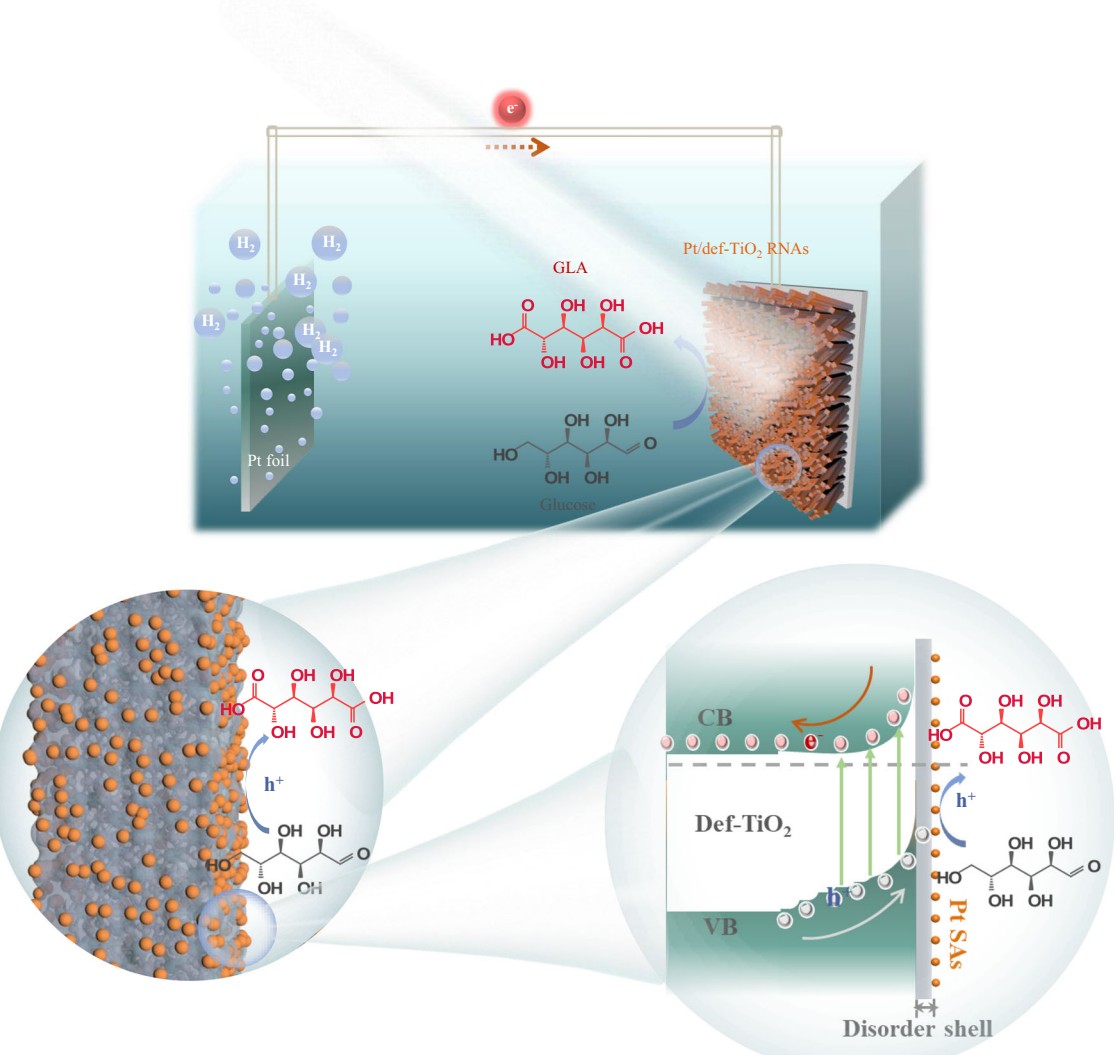

**Fig. 1 | Schematic illustration.** Selective PEC oxidation of glucose to GLA over the single-atom Pt decorated defective $TiO_2$ photoanode.

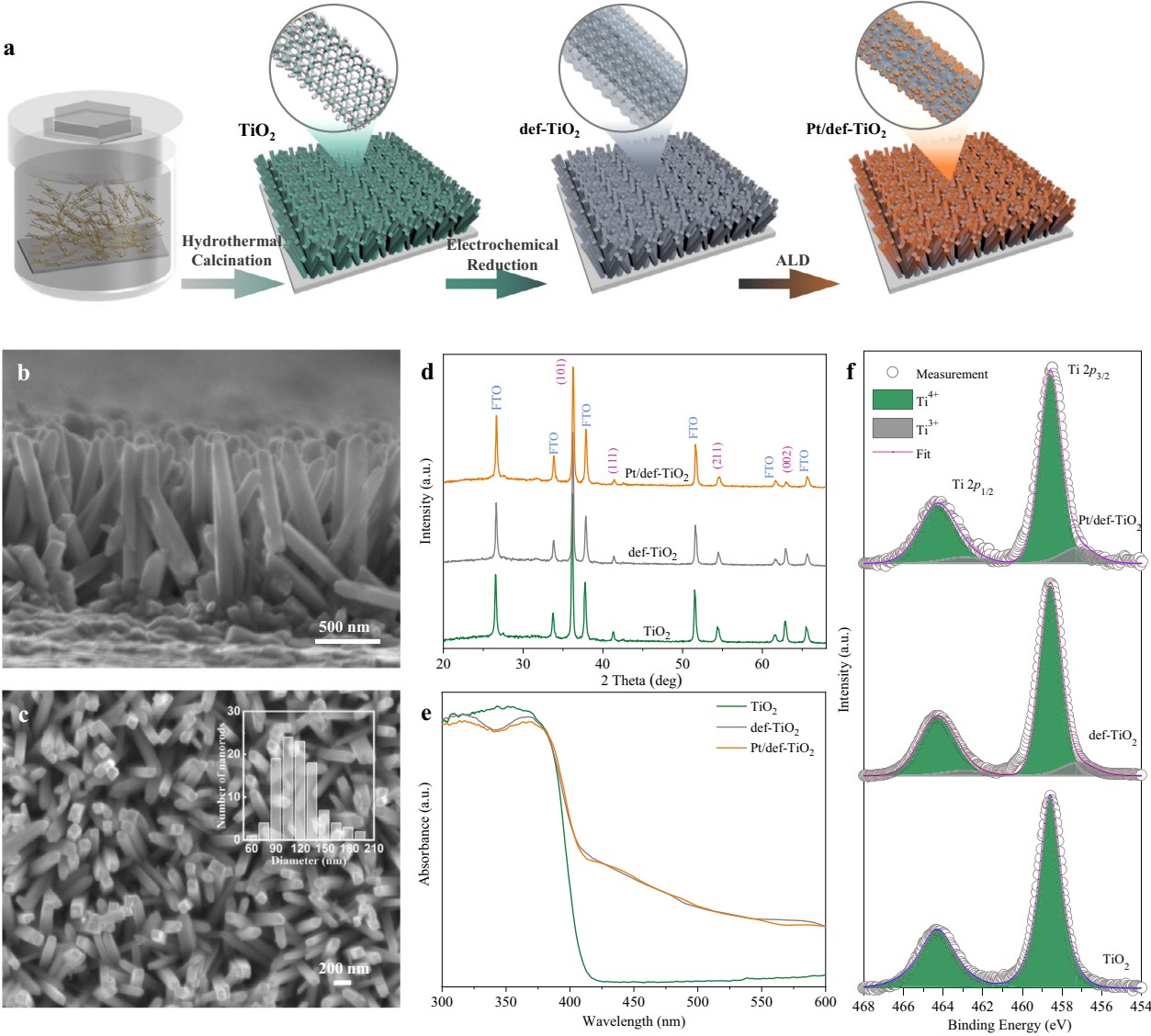

**Fig. 2 | Fabrication and characterization of the NRAs with Pt atoms. a** Schematic diagram of the fabricating process for the TiO$_2$, def-TiO$_2$, and Pt/def-TiO$_2$ NRAs. **b** Top-view and (**c**) side-view SEM images of the TiO$_2$ NRAs. Inset: the diameter distribution of the TiO$_2$ NRAs. **d** XRD patterns, (**e**) UV–vis absorption spectra of TiO$_2$, def-TiO$_2$, and Pt/def-TiO$_2$. **f** XPS spectra of Ti 2p for TiO$_2$ def-TiO$_2$, and Pt/def-TiO$_2$.

The defective TiO$_2$ (def-TiO$_2$) NRAs were fabricated by electrochemical reduction of TiO$_2$ NRAs in 0.5 M Na$_2$SO$_4$ electrolyte at −1.4 V$_{RHE}$ for 10 s. The single-atom Pt decorated def-TiO$_2$ (Pt/def-TiO$_2$) was obtained by atomic layer deposition (ALD) deposition of Pt single atoms (SAs) on def-TiO$_2$ at 200 °C for 10 cycles (Fig. 2a). The reduction treatment did not induce any phase transition in def-TiO$_2$ as suggested by the consistent XRD patterns of TiO$_2$ and def-TiO$_2$ (Fig. 2d). No additional diffraction peaks are observed in Pt/def-TiO$_2$, indicating that no obvious clusters of Pt metal and compounds are formed on def-TiO$_2$ NRAs. As shown in the diffuse reflectance UV–vis spectra (Fig. 2e), the TiO$_2$ NRAs present a typical rutile TiO$_2$ absorption with the edge at 420 nm, and its bandgap is determined to be 2.99 eV by the Tauc plots (Supplementary Fig. 1a). A slight red-shift is observed in the def-TiO$_2$ absorption spectrum, while the light absorption of def-TiO$_2$ is dramatically increased in the visible region, corresponding to a narrower bandgap of 2.70 eV (Supplementary Fig. 1a). This smaller bandgap can be attributed to the additional transitions caused by increased Ti$^{3+}$ states, oxygen vacancies, hydroxyl group, and other defects at different energy levels[19,20]. Further deposition of Pt atoms does not induce any obvious changes

in the absorption for def-TiO$_2$, due to the small loading or the limited size of Pt.

The influences of reduction treatment on chemical states were characterized by X-ray photoelectron spectroscopy (XPS) (Fig. 2f). Only two Ti 2p peaks centered at 464.3 and 458.6 eV with a splitting of 5.7 eV are observed in TiO$_2$, which are the typical Ti(IV) 2p$_{1/2}$ and Ti(IV) 2p$_{3/2}$ peaks in rutile TiO$_2$. After electrochemical reduction, additional two peaks appear beside the Ti(IV) 2p peaks, which are located at 462.9 and 457.2 eV with a splitting of 5.7 eV. These two peaks are thus attributed to Ti(III) 2p$_{1/2}$ and Ti(III) 2p$_{3/2}$. The emerging Ti$^{3+}$ signals verified that substantial oxygen vacancies are introduced into def-TiO$_2$ during the reduction process[19,21–23]. Additionally, the electron spin resonance (ESR) spectrum of def-TiO$_2$ shows a strong signal of trapped electrons in oxygen vacancies ($g$ = 2.002) (Supplementary Fig. 1b)[24], further confirming the introduction of oxygen vacancies by the reduction treatment. Numerous studies have revealed that the induced oxygen vacancies and disordered defects can induce an up-shift of the valance band, leading to a narrowed bandgap[22,25,26]. After deposition of Pt SAs, the Pt/def-TiO$_2$ NRAs show slightly stronger Ti(III) 2p signals (Fig. 2f) and ESR intensity (Supplementary Fig. 1b) than

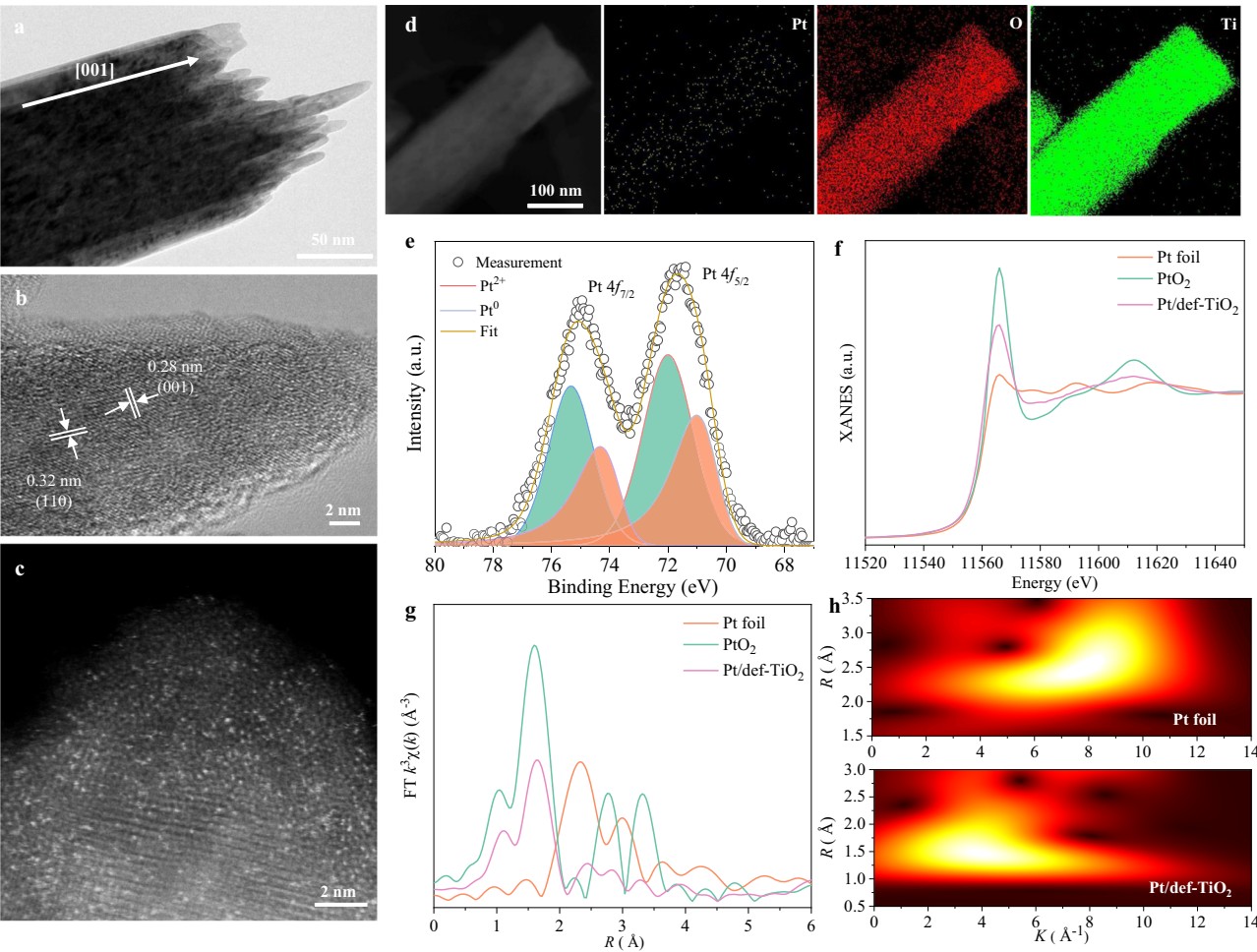

**Fig. 3 | Identification of single atoms on the defective TiO₂ nanorod. a** TEM and (**b**) HR-TEM images of Pt/def-TiO₂. **c** HAADF-STEM image of Pt/def-TiO₂ measured from a spherical aberration-corrected TEM. **d** STEM image and corresponding elemental (Ti, O, Pt) mappings of Pt/def-TiO₂. **e** XPS spectrum of Pt 4$f$ for Pt/def-TiO₂. **f** Pt L3-edge XANES spectra and (**g**) Fourier transformed EXAFS spectra of PtO₂, Pt foil, and Pt/def-TiO₂. **h** WT-EXAFS of Pt foil and Pt/def-TiO₂ at Pt edge.

def-TiO₂, probably due to the electron transfer from the Pt atoms to the def-TiO₂ support. This electron transfer between the def-TiO₂ support and the Pt SAs can lead to a strong metal-support interaction[27].

## Characterizations of single atoms

Single-atom Pt was also deposited on TiO₂ NRAs by ALD at 200 ℃ for 10 cycles (Pt/TiO₂) for comparison. After deposition of Pt atoms, Pt/TiO₂ shows no obvious change in the XRD pattern and UV-vis spectrum compared with TiO₂ NRAs (Supplementary Figs. 2, 3). The transmission electron microscopy (TEM) revealed that the diameter of the individual Pt/TiO₂ nanorod is about 120 nm (Supplementary Fig. 4a), consistent with the SEM results. High-resolution TEM (HRTEM) shows a single crystal structure of Pt/TiO₂ with lattice spacings of 0.28 and 0.32 nm, corresponding to the $d$-spacing of rutile TiO₂ in (101) and (110) planes, respectively (Supplementary Fig. 4b)[28]. Therefore, the axial direction of Pt/TiO₂ or Pt/def-TiO₂ nanorod is along the [001] direction (Fig. 3a). Following the XRD results, no nanoparticles are observed on the Pt/TiO₂ nanorod in the HRTEM image, suggesting that the adsorbed Pt ions were not agglomerated during the ALD process. The inductively coupled plasma optical emission spectroscopy (ICP-OES) affirmed the existence of Pt ions in the Pt/TiO₂ NRAs with a loading of 0.08 wt%. To have a direct view of the state of the Pt ions on the TiO₂ nanorod, the high-angle annular dark-field (HADDF-STEM) was employed to characterize Pt/TiO₂ (Supplementary Fig. 4c). The HADDF-STEM image shows that some isolated bright dots (indicated by the red dashed circles) can be distinguished from surrounding light

atoms of the nanorod, indicating the existence of single-atom Pt. Furthermore, the STEM image and corresponding energy dispersive spectroscopy (EDS) mapping of the Pt/TiO₂ nanorod demonstrate that the Pt atoms are evenly distributed over the whole nanorod (Supplementary Fig. 4d). After the reduction treatment, no obvious changes in the Pt/def-TiO₂ NA can be observed by the low-resolution TEM (Fig. 3a). However, in the HRTEM image (Fig. 3b), the Pt/def-TiO₂ NA shows a core-shell structure consisting of a crystalline core of rutile TiO₂ with a disordered shell layer. This disordered shell layer can provide more active sites for the onset MeCpPtMe₃ chemisorption, resulting in a huge increase in Pt loading. The loading is 0.7 wt% for Pt/def-TiO₂ under the same ALD conditions as Pt/TiO₂, and no agglomeration of adsorbed Pt ions is observed on the Pt/def-TiO₂ nanorod (Fig. 3b). The HADDF-STEM image further reveals a large amount of isolated Pt atoms on the surface of def-TiO₂ (Fig. 3c), suggesting that the disorder shell layer indeed provides sufficient active sites for single-atom Pt anchoring. The uniform distribution of Pt over the def-TiO₂ nanorod is confirmed by the STEM image and corresponding EDS mapping (Fig. 3d).

As shown in Fig. 3e, the high-resolution Pt 4$f$ spectrum can be deconvoluted into four peaks, where the peaks centered at 73.87 and 70.54 eV with a splitting of 3.33 eV are attributed to the Pt(0) 4$f_{7/2}$ and Pt(0) 4$f_{5/2}$, and the peaks located at 75.33 and 72.0 eV with a splitting of 3.33 eV are corresponding to Pt(II) 4$f_{7/2}$ and Pt(II) 4$f_{5/2}$. This suggests that a rich chemical environment of Pt SAs with different oxidation and charge states coexists on the surface of defective TiO₂[29].

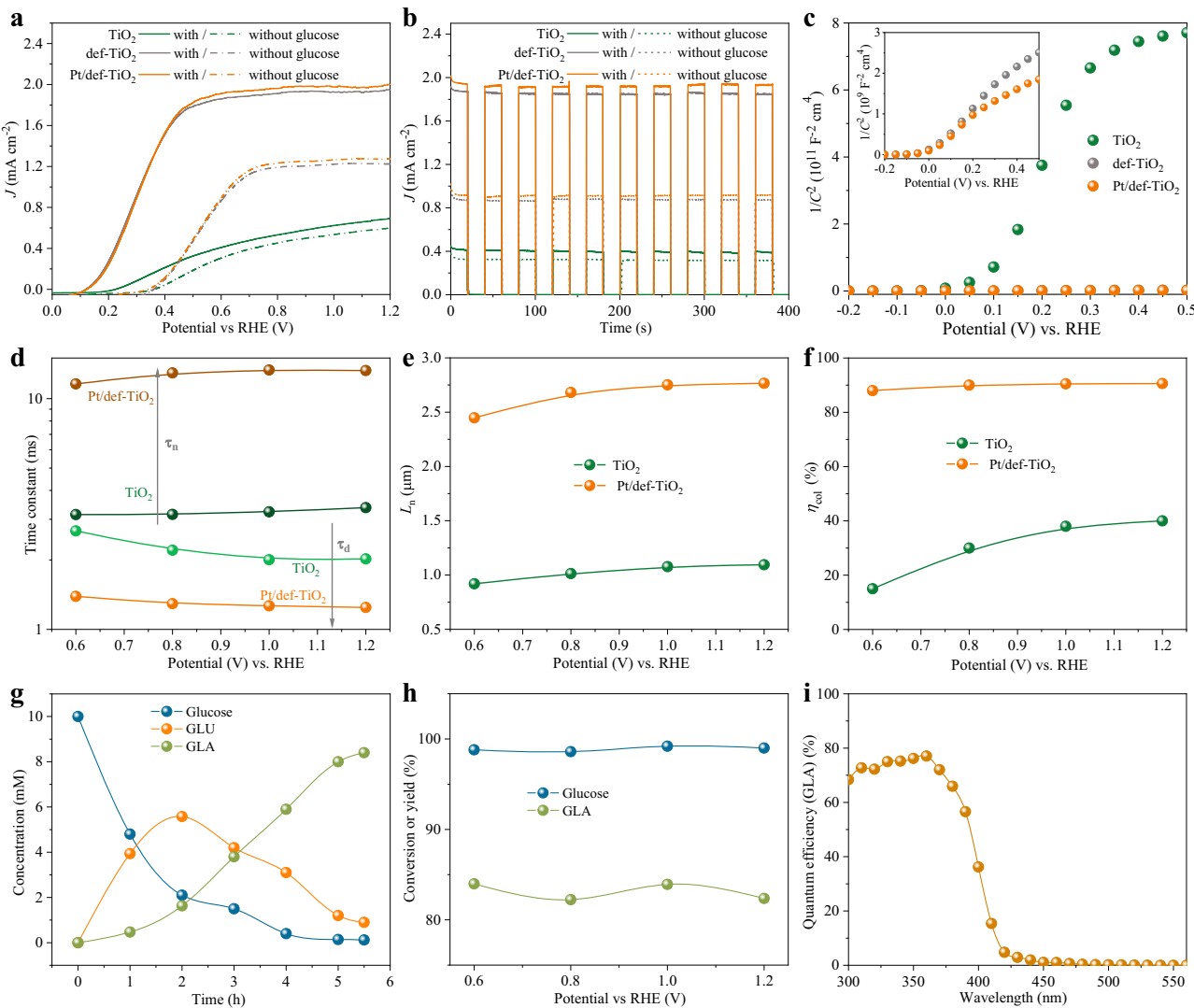

**Fig. 4 | PEC performance for glucose oxidation. a** LSV profiles of the TiO$_2$, def-TiO$_2$, and Pt/def-TiO$_2$ photoanodes for glucose oxidation measured in 1 M KOH with and without 10 mM glucose under AM 1.5 G, 100 mW cm$^{-2}$ illumination. **b** Transient photocurrent responses of the samples at 0.6 V$_{RHE}$ 1 M KOH with and without 10 mM glucose. **c** Mott–Schottky plots of TiO$_2$, def-TiO$_2$, and Pt/def-TiO$_2$. Inset: Magnified Mott-Schottky plots of def-TiO$_2$ and Pt/def-TiO$_2$. **d** Time constants of TiO$_2$ and Pt/def-TiO$_2$ photoanodes. **e** The corresponding electron diffusion length and (**f**) collection efficiency of TiO$_2$ and Pt/def-TiO$_2$ photoanodes versus applied potentials. **g** Time-dependent changes in the concentration of GLA, GLU, and glucose over the Pt/def-TiO$_2$ photoanode at 0.6 V$_{RHE}$. **h** Glucose conversation and GLA selectivity of the Pt/def-TiO$_2$ photoanode at different potentials. **i** Incident photo-to-GLA conversion efficiency spectrum of the Pt/def-TiO$_2$ photoanode at 0.6 V$_{RHE}$.

The existence of Pt(II) indicates that some Pt SAs are partially positively charged by electron transfer between the def-TiO$_2$ support and the Pt SAs, which leads to an enhanced metal-support interaction[27], consistent with the above Ti 2$p$ XPS (Fig. 2f) and ESR (Supplementary Fig. 1b) results. The X-ray absorption near-edge structure (XANES) spectra were further performed to clarify the electronic structure. The Pt L3-edge XANES spectra show that the white-line intensity of Pt/def-TiO$_2$ is lower than that of PtO$_2$ and higher than that of Pt foil (Fig. 3f), further evidencing the partial oxidation state of the single-atom Pt[27,30–33]. Fig. 3g shows the Fourier transformed extended X-ray absorption fine structure (FT-EXAFS) spectra of Pt/def-TiO$_2$ and references. Only one prominent peak at 1.6 Å is detected in Pt/def-TiO$_2$, attributed to the first shell of Pt-O scattering. Furthermore, compared with the FT-EXAFS spectrum of Pt foil reference, no higher shell of Pt-Pt scattering is observed in Pt/def-TiO$_2$, suggesting atomically dispersed Pt solely existing in Pt/def-TiO$_2$.

In addition, EXAFS wavelet-transform (WT) analysis that can strongly distinguish the backscattering atoms was performed to further consolidate the above inference. As shown in Fig. 3h, the WT

contour plots of Pt/def-TiO$_2$ and Pt foil reference at the Pt edge were put together for comparison. In the WT contour plots of the PtO$_2$ reference (Supplementary Fig. 5) and Pt foil reference (Fig. 3h), the first coordination shells show the intensity maximums at ~4 and ~8 Å$^{-1}$, ascribed to the Pt-O and Pt-Pt contributions, respectively. As illustrated in the WT contour plots of Pt/def-TiO$_2$ (Fig. 3h), only one Pt-O contribution at ~4 Å$^{-1}$ appears and no Pt-Pt contribution at ~8 Å$^{-1}$ is observed, further evidencing the atomically dispersed Pt atoms. Taken together with the abovementioned results, the Pt atoms are verified to be atomically dispersed in Pt/def-TiO$_2$.

## PEC performance

The PEC capabilities of the TiO$_2$ photoanodes with different reduction times and ALD cycles for glucose oxidation were first evaluated through the linear sweep voltammetry (LSV) experiments in 1 M KOH with 10 mM glucose under simulated sunlight (Supplementary Figs. 6, 7). After optimization, the photocurrent densities for glucose ($J_G$) and water oxidation ($J_W$) of the TiO$_2$, def-TiO$_2$, and Pt/def-TiO$_2$ photoanodes for glucose and water oxidation are shown in Fig. 4a.

**Table 1 | Photoelectrochemical oxidation of glucose at 0.6 $V_{RHE}$ in 1 M KOH with glucose**

| Entry | photoanode | $C_{glucose}$ (mM) | $J$ (mA cm$^{-2}$) | $t$ (h) | $X_{glucose}$ (%) | $Y_{CO2}$ (%) | $Y_{GLU}$ (%) | $Y_{GLA}$ (%) | $Y_{GLU+GLA}$ (%) |
|---|---|---|---|---|---|---|---|---|---|
| 1 | TiO$_2$ | 10 | 0.41 | 36 | >99.9 | 24.2 | 29.7 | - | 29.7 |
| 2 | Pt/TiO$_2$ | 10 | 0.46 | 36 | >99.9 | 18.4 | 11.2 | 26.4 | 37.6 |
| 3 | def-TiO$_2$-2 s | 10 | 0.78 | 20 | >99.9 | 7.2 | 42.4 | 8.1 | 50.5 |
| 4 | def-TiO$_2$-5 s | 10 | 1.42 | 10 | 98.9 | - | 69.8 | 10.2 | 81.0 |
| 5 | def-TiO$_2$ | 10 | 1.86 | 5.5 | 97.2 | - | 77.7 | 13.8 | 91.5 |
| 6 | def-TiO$_2$-15 s | 10 | 1.37 | 10 | 72.5 | - | 68.0 | 3.1 | 71.1 |
| 7 | Pt/def-TiO$_2$ | 10 | 1.91 | 5.5 | 98.8 | - | 9.2 | 84.3 | 93.5 |
| 8 | Pt/def-TiO$_2$ | 50 | 2.09 | 24 | 91.4 | - | 13.2 | 71.4 | 84.6 |
| 9 | Pt/def-TiO$_2$ | 100 | 2.31 | 48 | 87.6 | - | 14.4 | 63.5 | 77.9 |
| 10 | Pt/def-TiO$_2$-50 cycles | 10 | 1.74 | 6 | 97.2 | - | 47.5 | 48.4 | 95.9 |
| 11 | Pt/def-TiO$_2$-100 cycles | 10 | 1.41 | 8 | 98.3 | - | 57.1 | 32.9 | 95.0 |
| 12 | Pd/def-TiO$_2$ | 10 | 1.93 | 5.5 | 99.1 | - | 27.4 | 65.3 | 92.7 |
| 13 | Au/def-TiO$_2$ | 10 | 1.89 | 5.5 | 99.2 | - | 21.5 | 69.9 | 91.4 |

After reduction treatment, the def-TiO$_2$ photoanode shows a low applied potential to reach the saturated photocurrent ($E_{sat}$) for glucose oxidation (-0.4 $V_{RHE}$) and a high $J_G$ of 1.86 mA cm$^{-2}$ at 0.6 $V_{RHE}$. While the TiO$_2$ photoanode exhibits a much higher $E_{sat}$ for glucose oxidation (>0.8 $V_{RHE}$) and a much lower $J_G$ (0.41 mA cm$^{-2}$ at 0.6 $V_{RHE}$). This implies that moderate oxygen vacancies can efficiently promote the charge dynamics for PEC glucose oxidation[20,34]. The further deposition of Pt SAs on def-TiO$_2$ shows no effect on the $E_{sat}$ for glucose oxidation, but slightly enhances the $J_G$ of the Pt/def-TiO$_2$ photoanode (1.91 mA cm$^{-2}$ at 0.6 $V_{RHE}$), demonstrating the contribution of single-atom Pt to glucose oxidation. Without the addition of glucose in the electrolyte, both the $J_G$s from the def-TiO$_2$ and Pt/def-TiO$_2$ photoanodes were dramatically decreased, accompanied by obvious positive shifts of onset potentials and $E_{sat}$s (Fig. 4a). This is due to that the standard redox potential of glucose oxidation (0.05 V) is much lower than that of water oxidation (1.23 V)[35,36]. Importantly, the observed increments from $J_w$ to $J_G$ in the def-TiO$_2$ and Pt/def-TiO$_2$ photoanodes are much higher than those in the TiO$_2$ photoanode, suggesting that the reduction treatment can substantially suppress the water oxidation. The Ultraviolet Photoelectron Spectroscopy (UPS) results (Supplementary Fig. 8) exhibit a ~0.3 eV up-shit of the valance band maximum caused by the 10 s reduction treatment accounting for the suppressed water oxidation. Figure 4b shows the chopped photocurrent densities of the photoanodes at 0.6 $V_{RHE}$ with repeated on/off cycles correlated to the simulated solar light. All the photoanodes display stable photocurrents with fast and reproducible responses upon each cycle illumination.

To have a deep insight into the modulation of the defect structure on PEC glucose oxidation, the Mott−Schottky plots of the samples were collected in Fig. 4c and Supplementary Fig. 9. Compared to TiO$_2$, the def-TiO$_2$ and Pt/def-TiO$_2$ photoanodes show greatly enhanced carrier densities and negative shifts of the flat potentials (Supplementary Fig. 9). As a result, the depletion regions ($r_d$) of the def-TiO$_2$ and Pt/def-TiO$_2$ photoanodes are limited to 2.8–6.6 nm over the tested potentials (Supplementary Table 1). This limited $r_d$ will lead to a wider electron-conducting region for electron transportation[20,26,37], which was further evaluated by intensity-modulated photocurrent spectroscopy (IMPS) (Supplementary Fig. 10)[38–41]. As a result, the Pt/def-TiO$_2$ photoanode enables more efficient charge transport than the TiO$_2$ photoanode, consistent with a shorter electron transit time ($\tau_d$) (Fig. 4d). In the meantime, the Pt/def-TiO$_2$ photoanode also shows an enhanced electron lifetime ($\tau_n$), which was obtained from the small perturbation transient photocurrent measurements (Supplementary Fig. 11)[42]. The enhanced $\tau_n$ of the Pt/def-TiO$_2$ photoanode could be mainly attributed to its gradually elevated valance band maximum in the reduction shell induced by the reduction treatment. The electron collection efficiency and ($\eta_{col}$) diffusion length ($L_n$) can be

further obtained based on the equations of $L_n = L (\tau_n/\tau_t/2.35)^{1/2}$ and $\eta_{col} = 1 - \tau_t/\tau_n$[20,43]. The $L_n$ is thus determined to be around 1.0 μm for TiO$_2$ photoanode in the potential range of 0.6–1.2 $V_{RHE}$ (Fig. 4e), which is slightly shorter than the NRA thickness (1.3 μm). For the Pt/def-TiO$_2$ photoanode, $L_n$ is increased to around 2.5 μm due to the significantly improved electron conductivity. The longer $L_n$ of the Pt/def-TiO$_2$ photoanode achieves an efficient $\eta_{col}$ around 90% in the potential range of 0.6–1.2 $V_{RHE}$, while the smaller $L_n$ of the TiO$_2$ photoanode shows a $\eta_{col}$ less than 40% (Fig. 4f).

The products of the glucose oxidation from the Pt/def-TiO$_2$ photoanode at 0.6 $V_{RHE}$ under AM 1.5 G, 100 mW cm$^{-2}$ illumination were examined. The initial product was GLU, and it was then further oxidized into GLA (Fig. 4g). The glucose conversion ($X_{glucose}$) from the Pt/def-TiO$_2$ photoanode was 98.8% after a 5.5 h reaction, with GLU and GLA yields ($Y_{GLU}$ and $Y_{GLA}$) of 9.2 and 84.3%, respectively (Entry 7, Table 1), from which free-glucaric acid can be separated through ion exchange resin, separation by boronic acid affinity gel and azeotrope drying (Supplementary Fig. 12). Besides, the H$_2$ evolution at both Pt/def-TiO$_2$ photoanode and counter electrode was examined by online gas chromatography with Ar as carrier gas. The results show that H$_2$ was undetectable at the Pt/def-TiO$_2$ photoanode and only detected at the cathode electrode, meaning that the PEC oxidation of glucose on the photoanode could be a water-involved process. As shown in Supplementary Fig. 13, the faradaic efficiencies for H$_2$ evolution during 5.5 h PEC glucose oxidation are >99%, indicating no side reaction occurred on the counter electrode. The faradaic efficiencies for GLA and GLU are 86.8 and 3.1%, respectively. Besides, the $X_{glucose}$ and $Y_{GLA}$ from the Pt/def-TiO$_2$ photoanode at different applied potentials were also investigated (Fig. 4h). The Pt/def-TiO$_2$ photoanode shows almost identical $X_{glucose}$ and $Y_{GLA}$ at different applied potentials, due to the limited depletion regions obtained from the Mott−Schottky results. The incident photon-to-GLA efficiency spectrum shows that the Pt/def-TiO$_2$ photoanode can yield 70–80% photon-to-GLA conversion efficiencies in the UV region (Fig. 4i). Furthermore, the stability of the Pt/def-TiO$_2$ photoanode was examined by 5 runs of recycled $J$-$t$ test (27.5 h) of PEC glucose oxidation. As shown in Supplementary Fig. 14, the $J$-$t$ plot of the Pt/def-TiO$_2$ photoanode in the 1 first run shows that a stable photocurrent can be maintained for the first 3 h, while by prolonging the test to 5.5 h, a significant photocurrent attenuation was observed. This decreased photocurrent is mainly due to the depletion of glucose and GLU, which is verified by the recovered and reproduced photocurrents in the 2nd, 3rd, 4th, and 5th runs with the electrolyte refreshed every 5.5 h. Besides, after the 27.5 h test, no noticeable $J_G$ recession was observed, as presented in Supplementary Fig. 15a and b. These results suggest the good stability of the Pt/def-TiO$_2$ photoanode. In addition, after the repeated PEC reaction, no obvious

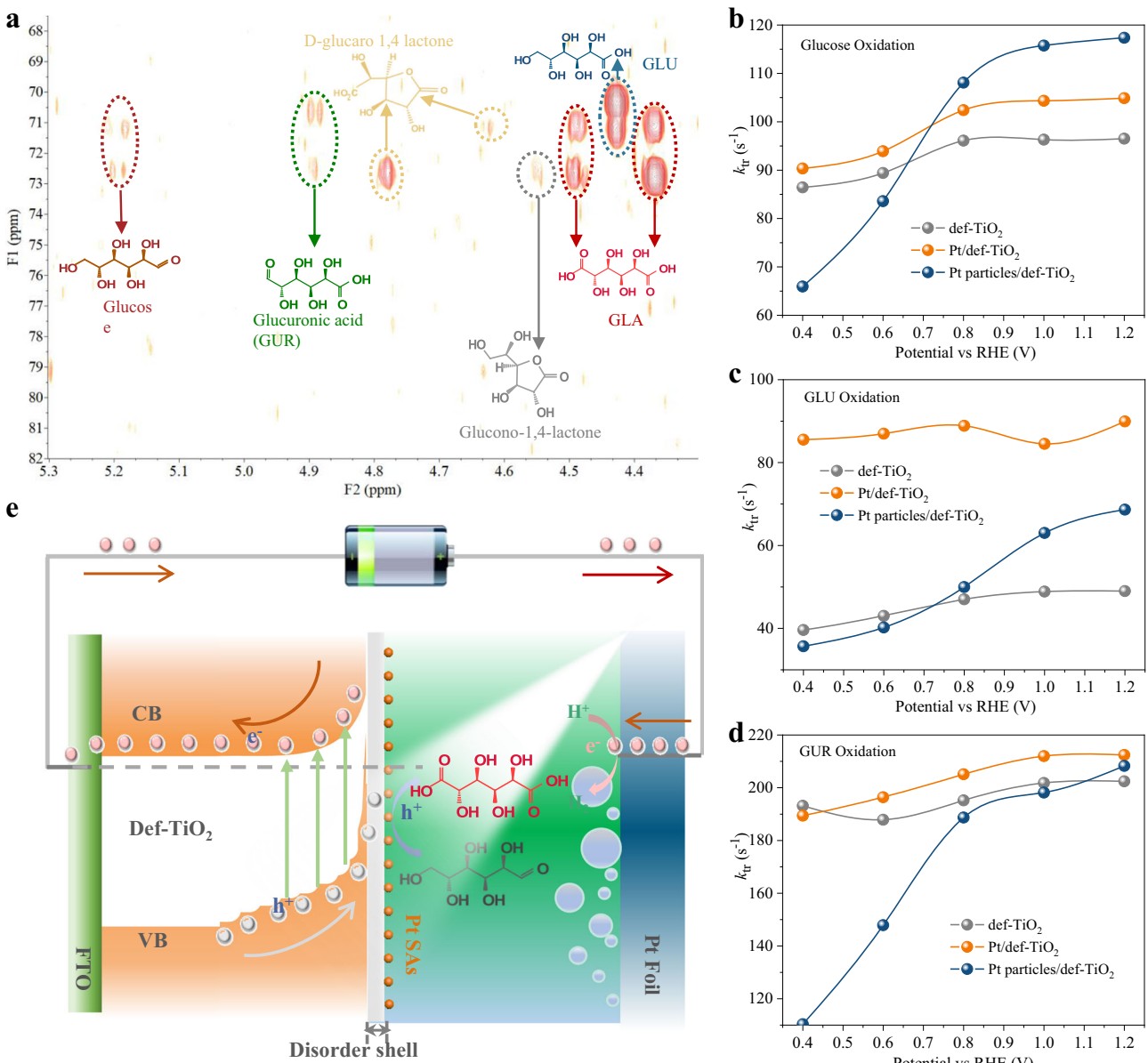

**Fig. 5 | The kinetics of PEC oxidation of glucose. a** $^{13}C - ^{1}H$ HMBC NMR spectrum of the products after 3 h oxidation over the Pt/def-TiO$_2$ photoanode under AM 1.5 simulated sunlight irradiation. Conditions: 1 M KOH deuteroxide solution with 10 mM glucose, at 0.6 V$_{RHE}$. The $k_t$ for (**b**) GLU, (**c**) GLA, and (**d**) GUR oxidation over the def-TiO$_2$, Pt/def-TiO$_2$, and Pt particles/def-TiO$_2$ photoanodes measured in 1 M KOH aqueous solution 10 mM glucose. **e** Energy diagram for the photo-induced charge transfer and transport based on the Pt/def-TiO$_2$ photoanode for PEC glucose oxidation.

changes were observed in the Ti 2$p$ and Pt 4$f$ XPS spectra of Pt/def-TiO$_2$ photoanode (Supplementary Fig. 15c, d), further revealing the good stability of the Pt/def-TiO$_2$ photoanode.

We also studied the influences of different photoanodes and different glucose concentrations ($C_{glucose}$) on the oxidation products (Entry 1–11, Table 1). The productions of GLU and CO$_2$ with $Y_{GLU}$ and $Y_{CO2}$ of ~29.7 and 24.2%, respectively, were observed by using the TiO$_2$ photoanode, and no GLA was detected. (Entry 1, Table 1). The CO$_2$ production is due to that the energy of photo-induced holes is too high for glucose oxidation, leading to cleavage of C-C bonds by most holes, which is supported by the diverse C$_1$-C$_5$ products detected by using TiO$_2$ as a photoanode in previous reports[44]. The product distributions of glucose oxidation over the TiO$_2$ photoanode from Entry 1, Table 1 demonstrate that various C$_1$-C$_6$ products were produced with only ~30% C$_6$ products (Supplementary Fig. 16), further confirming the role of high-energy holes in the cleavage of C-C bonds. Since the GLU is the initial product for GLA generation, such a

low concentration of GLU maybe not be sufficient for the production of GLA on the TiO$_2$ photoanode. The deposition of Pt SAs on TiO$_2$ didn't significantly enhance the $Y_{GLU+GLA}$, however, the GLA emerged with a yield of 26% due to the selectivity of Pt SAs to GLA (Entry 2, Table 1). Besides, the $Y_{CO2}$ on the Pt/TiO$_2$ photoanode is slightly decreased, probably due to that the promotion of Pt SAs to GLA leads to fewer holes involved in the cleavage of C-C bonds. With the reduction time increased from 2 s to 10 s, the defective TiO$_2$ photoanodes show increased $J_G$s and $Y_{GLU+GLA}$ (Entry 3–5, Table 1). The increased $J_G$s mainly result from the improved charge dynamics by the oxygen vacancies and the enhanced $Y_{GLU+GLA}$ is due to the decreased energy of photo-induced holes by the elevated valance band maximum (Supplementary Figs. 17, 18). A further careful study of reduction degree by the electron energy-loss spectroscopy shows that the def-TiO$_2$ nanorod possesses a ~21 nm-thick reduction shell and the content of oxygen vacancies is gradually increased from the inside to the surface, (Supplementary Fig. 19) corresponding

to a gradually increased O distortion. Consequently, the defective amorphous layer and gradually increased O distortion would lead to obviously elevated valence band tail states and gradually elevated valance band maximums in the reduction shell (Supplementary Fig. 20)[45], which are much higher than those determined by the absorption spectra. These gradually elevated valance band maximums and valence band tail states not only can promote charge carrier separation but also greatly reduce the energy of holes, suppressing the C-C bonds cleavage. Therefore, with the increase of reduction time, the $Y_{CO_2}$s over the defective photoanodes are greatly decreased, and no detectable $CO_2$ is observed after 2 s reduction (Entry 4–11 Table 1). Benefiting from this, the defective $TiO_2$ photoanode with 10 s reduction (def-$TiO_2$) shows a high photocurrent and a high $Y_{GLU+GLA}$ of 91.5% (Entry 5, Table 1), however, its $Y_{GLA}$ is still low (13.8%). While the further deposition of Pt SAs can greatly increase the $Y_{GLA}$ to 84.3% (Entry 7, Table 1). TEM results show that the thicknesses of the disorder shell vary from ~0.5 to ~4 nm with a reduction time from 2 s to 10 s (Supplementary Fig. 21). Further prolonging the reduction time to 15 s, the thickness of the disorder shell is increased to ~8 nm (Supplementary Fig. 21), resulting in an over reduction of the photoanode for glucose oxidation.

The defective photoanode with 15 s reduction exhibits excessive oxygen vacancies acting as the recombination centers and insufficient oxidation capacity by the over elevated valance band maximum (Supplementary Figs. 17, 18). $X_{glucose}$ and $J_G$ are consequently decreased (Entry 6, Table 1). When the ALD was prolonged to 50 cycles some Pt particles were formed on def-$TiO_2$ (Supplementary Fig. 22a), while dense Pt particles with a diameter of ~2 nm were formed on def-$TiO_2$ after 100 cycles (Supplementary Fig. 22b). The formed Pt particles are metallic Pt as suggested by the XPS spectrum (Supplementary Fig. 23). The increased metallic Pt particles can substantially decrease the $J_G$ (Entry 10–11, Table 1) due to that the oxygen vacancies between the disorder shell and the Pt particles act as the recombination centers. Importantly, with the increase of metallic Pt particles, the photoanodes show significantly decreased $Y_{GLA}$, implying the essence of Pt SAs for the selective production of GLA. These results demonstrate the key role of the interaction between the disordered $TiO_2$ and Pt SAs in the selectivity of glucose to GLA. Besides, the Au and Pd SAs were also successfully deposited on the def-$TiO_2$ (Au/def-$TiO_2$ and Pd/def-$TiO_2$) photoanode through an immersion method to investigate their PEC performances for glucose oxidation (Supplementary Fig. 24). As shown in Entry 12 and 13, Table 1, the Pd/def-$TiO_2$ and Au/def-$TiO_2$ photoanodes also show significantly enhanced $Y_{GLA}$ of 65.3 and 69.9%, respectively, compared with the def-$TiO_2$ photoanodes, implying the facilitations of Au and Pd SAs on the conversion of glucose to GLA. However, the $Y_{GLA}$s from the Pd/def-$TiO_2$ and Au/def-$TiO_2$ photoanodes are much lower than that from the Pt/def-$TiO_2$ photoanode. Therefore, although the Pt SA is not the unique metal to promote the selective oxidation of glucose to GLA, it is the most active single atom for the selective GLA generation.

## Reaction mechanism and path

Finally, the reaction pathway and mechanism over the Pt/def-$TiO_2$ photoanodes were investigated by the 2D-HMBC NMR and rate constant of charge transfer ($k_t$) from the IMPS. To obtain the 2D-HMBC NMR, the PEC oxidation of glucose was conducted in 20 ml deuteroxide as the solution with 1 M KOH and 10 mM glucose, and the final solution was adjusted to acidity with $H_2SO_4$ prior to NMR analysis. The possible glucose decomposition in 1 M KOH at 20 °C was investigated by 2D-HMBC NMR. No obvious changes were observed in the 2D-HMBC NMR spectra of the solutions before and after keeping for 10 h (Supplementary Fig. 25), suggesting no significant self-decomposition of glucose. This is consistent with previous studies of glucose oxidation in alkali solutions[7,46,47]. As shown in the 2D-HMBC NMR spectrum after 1 h oxidation (Supplementary Fig. 26), glucose, GLU, and some of

their lactonization derivatives were observed. Only one kind of the lactonization derivative of GLU, i.e. D-glucono 1,4 lactone, was detected. After 3 h oxidation, except for the GLU and D-glucono 1,4 lactone, the GLA, the lactonization derivative of GLA (D-glucaro 1,4 lactone), and L-guluronic acid (GUR) emerged (Fig. 5a). It should be noted that the D-glucono 1,4 lactone is substantially declined and a small amount of D-glucaro 1,4 lactone is generated, indicating lactonization derivatives should also participate in a reaction. When oxidation duration was prolonged to 5.5 h, most of the glucose was converted into GLA equilibrium with a tiny amount of, D-glucaro 1,4 lactone, and a small quantity of GLU resided in the electrolyte (Supplementary Fig. 27). Therefore, the oxidation of glucose follows the path from glucose to GLU, then to GUR, and finally to GLA, meanwhile equilibrium with their lactonization derivatives (Supplementary Fig. 28), and their lactonization derivatives also should participate in the series of oxidation reactions. Furthermore, the Pt/def-$TiO_2$ NRAs were separately immersed into the electrolytes of 1 M KOH with 10 mM glucose, 1 M KOH with 10 mM GLU, and 1 M KOH with 10 mM GUR without any applied voltage. After 5 h illumination of simulated sunlight at 20 °C, the electrolytes were analyzed by HPLC and the results show that trace amounts of GLA were detected in the electrolytes, suggesting that the oxidation of GLU to GLA proceeds over Pt SAs is indeed a PEC reaction.

To have a deep insight into the mechanism of serial oxidation of glucose to GLA, the $k_t$ for glucose, GLU, and GUR oxidation over the def-$TiO_2$, Pt/def-$TiO_2$, and Pt/def-$TiO_2$ with 100 cycles (Pt particles/def-$TiO_2$) photoanodes were examined (Fig. 5b–d). The corresponding rate constant of surface recombination ($k_{rec}$) and surface charge transfer efficiency ($\eta_{tran}$) for glucose, GLU, and GUR oxidation are computed, respectively (Supplementary Figs. 29–31). The $k_t$ for glucose, GLU, and GUR oxidation over the def-$TiO_2$ photoanode are around 95, 45, and 190 s$^{-1}$, respectively. The much lower $k_t$ for GLU but much higher $k_t$ for GUR indicates that the oxidation from GLU to GUR is the rate-limiting step for the selective oxidation of glucose to GLA. Additionally, the def-$TiO_2$ photoanode also shows much lower $\eta_{tran}$ for GLU oxidation than those for glucose and GUR oxidation, further evidencing the rate-limiting step of GLU oxidation (Supplementary Figs. 29–31). In addition, the much higher $k_t$ of all the samples for GUR indicates the GUR can be easily converted to GLA. This can explain why only a small amount of GUR was detected during the whole oxidation process.

The Pt SAs has an all-round effect on the promotion of the oxidation of aldehyde groups from glucose and alcohol hydroxyl groups from GLA, evidenced by their enhanced $k_t$s on the Pt/def-$TiO_2$ photoanode (Fig. 5b, c). Importantly, the much lower $k_t$ for GLU is significantly increased to around 90 s$^{-1}$, which can greatly accelerate the GLA conversion. This reveals that the selectivity of glucose to GLA over the Pt/def-$TiO_2$ photoanode mainly stems from the accelerated kinetics of the GLU oxidation by the Pt SAs. Although the metallic Pt particles can accelerate the conversion of glucose to GLU (Fig. 5c), GLU oxidation is still limited, resulting in a low GLA selectivity. Besides, LSV curves for GLU, GUR, and GLA oxidation over the $TiO_2$, def-$TiO_2$, and Pt/def-$TiO_2$ photoanodes were tested with and without illumination (Supplementary Fig. 32), which further confirmed the PEC reaction for glucose oxidation and fast kinetics for GUR oxidation. These LSV results also reveal the accelerated GLU oxidation over the Pt SAs and sluggish GLA oxidation, consequently resulting in a high selectivity of glucose to GLA over the Pt/def-$TiO_2$ photoanode. Based on the above results, both the defective $TiO_2$ and the Pt SAs are crucial for the selective oxidation of glucose to GLA (Fig. 5e). The def-$TiO_2$ with a large electron conduction region and large surface band bending can greatly enhance the charge carrier extraction. The extracted electrons are transported to the counter electrode for $H_2$ production, while the extracted holes are regulated to the appropriate energy for glucose oxidation by the valance band structure of the defective $TiO_2$. Finally, the holes are collected by the glucose through the Pt SAs, realizing selective oxidation to GLA.

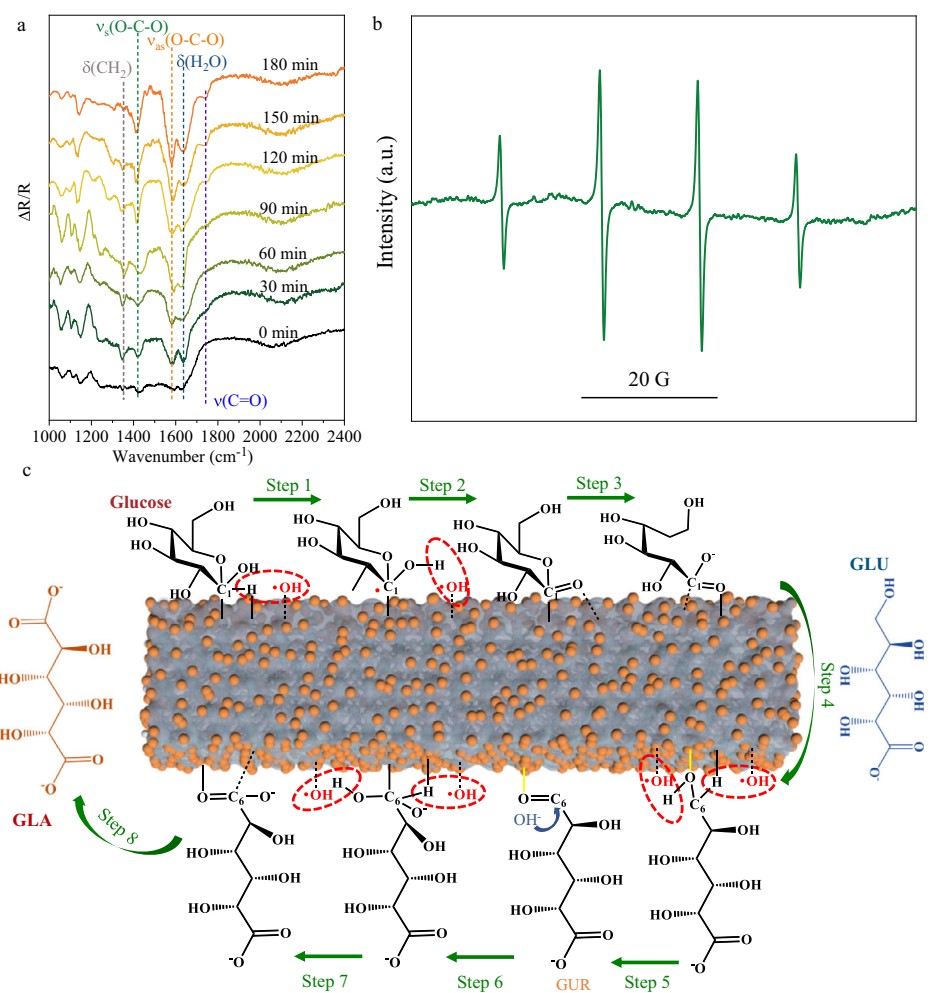

**Fig. 6 | Possible reaction pathway from glucose to GLA. a** In-situ Fourier transform infrared spectra for PEC glucose conversion on the Pt/def-TiO$_2$ photoanode in 1 M KOH with 10 mM glucose. **b** EPR detection of •OH using DMPO as a spin-trapping agent under illumination with Pt/def-TiO$_2$ in 1 M KOH. **c** Schematic illustration of the possible pathway for the PEC oxidation of glucose to GLU and GLA over the Pt/def-TiO$_2$ photoanode.

To elucidate the reaction pathway of glucose oxidation, electron paramagnetic resonance (EPR) spectroscopy and in-situ Fourier transform infrared spectra (FTIR) measurements were carried out to monitor the reaction intermediates. Figure 6a shows the in-situ FTIR of the Pt/def-TiO$_2$ photoanode in the reaction of PEC glucose oxidation recorded at 0.6 V$_{RHE}$ from 0 to 180 min. The negative characteristic IR bands located at -1350, 1415, 1585, 1640, and 1740 cm$^{-1}$ are assigned to δ(CH$_2$) in the C6-position of gluconate anion and glucose, symmetric ν$_s$(O−C−O) of COO$^-$ function in gluconate (C1-position), L-glucuronate (C1-position) and glucarate (C1-position and C6-position), asymmetric stretching vibration ν$_{as}$(O−C−O) of COO$^-$ function, δ(H$_2$O) due to interfacial water absorbed on the photoanode, and ν(C=O) in gluconolactone or L-glucuronate (C6-position), respectively[48–53]. The ν$_s$(O−C−O) and ν$_{as}$(O−C−O) IR bands show obvious dependence on PEC duration, suggesting the continuous generation of the COO$^-$ function. Remarkably, no obvious C-C breaking occurred during the PEC glucose oxidation over the Pt/def-TiO$_2$ photoanode as revealed by the undetected C−C bond cleavage compounds (CO: 1900−2100 cm$^{-1}$, carbonate: 1396 cm$^{-1}$, CO$_2$: 2340−2350 cm$^{-1}$)[48]. The stable δ(H$_2$O) IR bands during the PEC glucose oxidation indicate the involvement of water in the reaction. Further EPR spectrum exhibits quartet signals with an intensity ratio of 1:2:2:1, assigned to the DMPO−•OH adduct (Fig. 6b)[54]. This means that the absorbed water molecules on the catalyst are oxidized by the holes to form •OH radicals. Meanwhile, •OH radicals and holes

will separately attack the H atom of the C1-H, and the C1-O-H bonds, resulting in the formation of C$_1$-OO$^-$ groups[48,49]. Besides, the peak intensities of δ(CH$_2$) remain almost unchanged until 90 min and then decrease gradually with increased ν(C=O) IR bonds, suggesting that the oxidation process mainly occurred on the C6 position after 90 min. These results suggest that the transformation of glucose into GLA is achieved through the intermediate production of GLU and GUR.

A reaction mechanism for the PEC oxidation of glucose to GLA on the Pt/def-TiO$_2$ photoanode can therefore be proposed (Fig. 6c). The holes from the photoanode will react with the absorbed water to form absorbed •OH radicals[52,55], which can abstract the H atoms of the C1-H and C1-O-H bonds from absorbed C1 of glucose on the photoanode forming C1=O bond in step 2[49,56]. Subsequently, the C5-O bond is split by hydrolysis, and then the GLU is formed through desorption from the photoanode[48,49]. Previous study has shown that highly dispersed Pt catalysts are capable of selectively activating primary alcohols for further oxidation[57]. Therefore, our Pt SAs on the photoanode are proposed to selectively adsorb primary alcohols from the C6 position. After the co-adsorption of C6-OH on the Pt SAs and C6 on the photoanode (step 4), the absorbed •OH radicals will react with the H atoms of C6-OH and C6-H, separately, to form a C6=O bond (step 5), where some products desorb from the catalyst to generate GUR and others will adjust their adsorption states for the subsequent reaction. During step 6, with the holes and hydroxyl groups, the C6=O bond will be

activated and a new C6-OH bond will be formed[58]. Subsequently, the groups from the C6 position will follow the same reaction pathway as the C1 position to yield GLA, eventually. Therefore, both the Pt SAs and absorbed •OH radicals can play key roles in the selective PEC oxidation of glucose to GLA.

In summary, a PEC method was developed to selectively convert glucose to high value-added GLA. The Pt/def-TiO$_2$ photoanode derived from single-atom Pt anchored on defective rutile TiO$_2$ nanorod arrays shows a high yield of GLA. The results suggest that the interaction between the disordered TiO$_2$ and single-atom Pt is crucial in the selective production of GLA. The defective structure is revealed to facilitate charge separation and transport, leading to a huge increase in $J_G$. Importantly, the energy of valance band holes can be modulated by the defect structure, realizing the optimized yield of C$_6$ products. The selectivity of the GLA can be regulated by the Pt SAs through the acceleration of GLU oxidation. Therefore, a high of 84.3% for GLA at 0.6 V$_{RHE}$ under simulated sunlight illumination was achieved on the Pt/def-TiO$_2$ photoanode. This PEC strategy for selective oxidation of organic compounds reveals a new strategy for the utilization of biomass feedstocks.

## Methods

### Growth of Rutile TiO$_2$ NRAs
The TiO$_2$ NRAs were grown on FTO substrates (7 Ω per square, Film: 700 nm, NSG Company) by a hydrothermal method[18]. In a typical synthesis, a tetrabutyl titanate precursor solution was first obtained by slowly mixing 0.7 ml tetrabutyl titanate (Alfa Aesar, 99%), 21 ml concentrated HCl (Alfa Aesar, 37%), and 30 ml deionized water. Before use, the FTO was ultrasonically cleaned with ethanol, the mixture of acetone, isopropanol, and water (1:1:1), acetone in sequence, and kept in the acetone solution. Subsequently, the precursor solution was transferred into a 50 mL sealed Teflon reactor with four pieces of cleaned FTO, which was kept at 180 °C for 4 h. Finally, the FTO substrates with as-synthesized NRAs were rinsed with deionized water, dried in air, and annealed in air at 500 °C for 1 h to obtain TiO$_2$ NRAs.

### Fabrication of defective TiO$_2$ NRAs
The defective NRAs were obtained through an electrochemical reduction method[59]. In detail, a conventional three-electrode system was connected to a potentiostat electrochemical workstation (CHI 760E), where the TiO$_2$ NRAs on the FTO, an Ag/AgCl (KCl saturated), and a Pt wire were used as the working, reference, and counter electrodes, respectively. The electrochemical reduction was carried out in 0.5 M Na$_2$SO$_4$ (pH = 6.8) electrolyte at an applied potential of −1.4 V vs reversible hydrogen electrode (V$_{RHE}$) for 2, 5, 10, and 15 s to obtain defective TiO$_2$ NRAs.

### Fabrication of single-atom Pt, Pd, and Au on TiO$_2$ and def-TiO$_2$
The single-atom Pt was deposited on the TiO$_2$ and def-TiO$_2$ NRAs through an ALD method. During the ALD process, the trimethyl(methylcyclopentadienyl)-platinum (IV) (MeCpPtMe$_3$), O$_3$, and high purity N$_2$ were used as precursors, reaction gas, and carrier gas, respectively. The TiO$_2$ and def-TiO$_2$ on the FTO substrates were placed into the ALD chamber. The MeCpPtMe$_3$ precursor and gas lines were kept at 70 and 120 °C to supply continuous precursor vapor to the chamber. The Pt/TiO$_2$ and Pt/def-TiO$_2$ were obtained by the ALD at the temperature of 200 °C for 10 cycles. 50 and 100 cycles of the ALD were also conducted on the def-TiO$_2$ NRAs for comparison.

The single-atom Pd and Au were deposited on the def-TiO$_2$ NRAs through an immersion method. In detail, the def-TiO$_2$ NRAs were immersed into 1 mmol/L HAuCl$_4$ and H$_2$PdCl$_4$ solutions for 30 mins, respectively. Subsequently, the NRAs on the FTO substrates were annealed at 200 °C for 0.5 h in air to obtain the Pd/def-TiO$_2$ and Au/def-TiO$_2$ photoanodes.

## Data availability
The data that support the findings of this study are available within the article and its Supplementary Information files. All other relevant data supporting the findings of this study are available from the corresponding authors upon request.

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

## Acknowledgements

Authors acknowledge financial support by the National Research Foundation, Singapore, and A*STAR (Agency for Science, Technology and Research) under its LCER FI program Award No U2102d2002, and NUS R&G Postdoc Fellowship Program.

## Author contributions

Z.T., Y.D., and W.C. conceived and designed the experiments. Z.T., Y.D., M.W., X.D., R.J., S.X., B.C., and Y.X. performed the experiments. Z.T., Y.D., J.C., Y.L., and H.Y. analyzed the data. T.Z., X.C., Y.D., and Y.L. wrote the manuscript. All authors discussed the results and commented on the manuscript.

## Competing interests

The authors declare no competing interests.
