## [Peer Review File · Nature Communications]

Selective Photoelectrochemical Oxidation of Glucose to Glucaric Acid by Single Atom Pt Decorated Defective TiO₂REVIEWER COMMENTS

Reviewer #1 (Remarks to the Author):

This work demonstrates the photoelectrochemical approach for selective oxidation of glucose to value-added glucaric acid and Pt anchored defective TiO₂ nanorod arrays have been synthesized as photoanode. High performances, such as photocurrent density and glucaric acid have been obtained. The high performances are mainly attributed to the defects of TiO₂ and Pt loading. Although intensive work has been done, some scientific issues remain unclear. I cannot recommend its publication at the present form.

1. C-C bond cleavage has been found in the oxidation of glucose under the catalysis of different materials at different conditions, which has been simply attributed to the high energy of photo-induced holes. What is the differences of photo-induced holes that leading to the variation of YGLU+GLA?
2. The selective oxidation of glucose has been reported by many approaches, such as electrolysis with high current density and high yield of glucaric acid. What is the advantages of this approach?
3. Is there any beside products, such as hydrogen?
4. The produced glucaric acid will react with the KOH electrolyte, therefore the real product is corresponding salt. Can it be separated?

Reviewer #2 (Remarks to the Author):

This manuscript describes an interesting study on using Pt/def-TiO₂ photoanode to selectively oxidize glucose to gluconic acid. This work may proved a new PEC strategy for the utilization of biomass feedstocks. The strategy presented as well as the experimental results are interesting and novel. However, the authors were not able to elucidate well the mechanisms of selective oxidation of glucose to glucodipic acid and intermediates produced during oxidation (non-intermediate products) when Pt SAs and def-TiO₂ coexist. Meanwhile, the rigor of the current study can be further improved in this manuscript. The current manuscript needs to undergo major revisions before being considered for publication:

- 1) The authors reported an 84.3 % yield of glucaric acid, what is the faradaic efficiency (current efficiency) to GLA? Is there be some other gaseous products, such as O₂, CO₂, CO, etc? Is there H₂O₂ in the liquid product? The authors should report faradaic efficiency for all products, and the calculation method can ref to Adv. Mater. 34, 2201594 (2022).
- 2) The authors believe that the TiO₂ NRAs established a low yield of GLA is due to the energy of photo-induced holes is too high for glucose oxidation, leading to cleavage of C-C bonds by most holes. The authors should provide product distributions to verify this claim.
- 3) The authors claimed that the Pt/def-TiO₂ photoanode can maintain its activity during its 5 runs of recycled (27.5 h). Long-term photocurrent or 5 runs of recycles photocurrent data is necessary to provide for a more rigorous description.
- 4) As far as I know, in the photocatalytic glucose oxidation reaction, some photocatalysts with less

positive valence bands can also break the C-C bonds of glucose to form low-carbon chemicals such as lactic acid (eg. ACS Catal. 12, 11206–11215 (2022); CCS Chem. doi: 10.31635/ccschem.022.202202213 (2022)). Therefore, why Pt/def-TiO₂ can selectively oxidize glucose to GLA but not other C1-C5 chemicals?

5) The authors proved that Pt SAs on def-TiO₂ can significantly enhance the *k_t* for GLU. In fact, Pt can not only promote the oxidation of alcohol hydroxyl groups, but also promote the oxidation of aldehyde groups according to previous reports (eg. Angew. Chem. Int. Ed. 60, 22908–22914 (2021)). Therefore, the promotion of the oxidation of glucose to GLA by Pt SAs may have an all-round effect.

6) Is Pt a unique metal to promote the selective oxidation of glucose to GLA? What is the real catalytic site of glucose oxidation?

7) The authors provide proposed possible reaction pathway from glucose to GLA, but it is slightly simple. And how H₂O is involved in the oxidation of glucose, and the charge transfer at each step have not been elucidated. Can the authors obtain some more accurate intermediates their evolution and transformation process through some in situ characterization, such as ESR and DRIFTS.

Other minor revisons:

1) Statements and abbreviations for nanorod arrays are not uniform in this manuscript, and there are spelling errors such as in lines 82 and 83.

2) In this manuscript, in line 148, the growth drection of TiO₂ NRAs is described, but Fig. 3a is the growth drection of Pt/def-TiO₂ NRAs.

3) In the SI, in line 149-150, “When compared to the Pt/def-TiO₂ photoanodes, the Mott-Schottky changes in the Pt/def-TiO₂ photoanodes are not obvious” should be corrected to “When compared to the def-TiO₂ photoanodes, the Mott.Schottky changes in the Pt/def-TiO₂ photoanodes are not obvious”.

4) The Methods section should be described in more detail, e.g. photoanode size, cell volume, electrolyte volume.

Reviewer #3 (Remarks to the Author):

The paper focuses on the selective photoelectrochemical oxidation of glucose using single-atom Pt decorated defective TiO₂ (Pt/def-TiO₂) to obtain the high value-added glucaric acid. The overall quality and novelty of this paper may be suitable for your esteemed journal. However, it does not sufficiently discuss the reaction mechanism and the role of Pt. The authors should address the following points:

1. The amount of CO₂ under low GLA+GLU yield conditions in glucose oxidation should be added.

2. The LSV curves for GLU, GUR, and GLA oxidation should be added using Pt, TiO₂, def-TiO₂, and Pt/def-TiO₂ (photo)electrodes under light and dark conditions, respectively.

3. In Fig. 5, the authors should add (1) oxidation rates at low applied bias, including no applied bias conditions, (2) faradaic efficiencies of the oxidation products, and (3) the amount of CO₂ production.

4. The role of Pt is unclear. If oxidation of GLU to GLA proceeds over Pt, is it a photoelectrochemical reaction? In particular, for the oxidation of intermediates, is it possible that the reaction proceeds photo or thermally without needing an applied voltage? The authors need to discuss these points further.

Responses to Reviewers

Reviewer 1:

Comment 1: C-C bond cleavage has been found in the oxidation of glucose under the catalysis of different materials at different conditions, which has been simply attributed to the high energy of photo-induced holes. What is the differences of photo-induced holes that leading to the variation of $Y_{\text{GLU+GLA}}$?

Response: We are grateful for the time and energy the reviewer expended on our behalf. As shown in **Supplementary Fig. 17**, as the introduction of oxygen vacancies increases, the bandgap of the defective TiO_2 gradually decreases.^{1,2} Further UPS results (**Supplementary Fig. 8**) reveal that the reduction treatment can induce an elevated valance band maximum, resulting in decreases in the energy of photo-induced holes. Therefore, with the reduction time increased from 2s to 10s, C-C bonds cleavage is suppressed, leading to increased $Y_{\text{GLU+GLA}}$. However, the defective photoanode with 15 s reduction exhibits excessive oxygen vacancies acting as the recombination centers and insufficient oxidation capacity by the over elevated valance band maximum (**Supplementary Fig. 16 and 17**), leading to a decreased X_{glucose} accompanied by reduced $Y_{\text{GLU+GLA}}$. A further careful study of reduction degree by the electron energy-loss spectroscopy shows that the def- TiO_2 nanorod possesses a ~21 nm-thick reduction shell and the reduction degree is gradually increased from the inside to the surface (**Fig. R1**), corresponding to a gradually increased O distortion. Consequently, the defective amorphous layer and gradually increased O distortion would lead to obviously elevated valance band tail states and gradually elevated valance band maximums in the reduction shell (**Fig. R2**),³ which are much higher than those determined by the absorption spectra. These gradually elevated valance band maximums and valance band tail states can not only promote charge carrier separation but also greatly reduce the energy of holes, suppressing the C-C bonds cleavage. Therefore, with the increase of reduction time, the $Y_{\text{CO}_2\text{S}}$ over the defective photoanodes are greatly decreased, and no detectable CO_2 is observed after 2s reduction (Entry 4-11 **Table 1**).

Supplementary Fig. 17 (a) UV-vis absorption spectra and (b) corresponding tauc plots of defective TiO_2 photoanodes with different reduction times. The bandgaps of defective TiO_2 photoanodes decrease over the reduction time, and up-shift VB maximums should account for the decreased bandgaps.

Supplementary Fig. 8 Band alignment between TiO_2 and Pt/def-TiO_2 . (a) and (b) UPS spectra of TiO_2 and Pt/def-TiO_2 . (c) Band structure diagram of TiO_2 and Pt/def-TiO_2 based on the UPS results.

Supplementary Fig. 16 ESR spectra of defective TiO₂ photoanodes with different reduction times. The concentration of the oxygen vacancies in the defective TiO₂ photoanodes increases over the reduction time.

Fig. R1 (a) STEM image of a def-TiO₂ nanorod with the probing path shown by the red circles and (b) the corresponding EELS spectra of titanium L_{2,3} edge. The pitch between the two neighboring spectra is ~ 3.5 nm.

Fig. R2 Energy band structure of the def-TiO₂ photoanode and corresponding Energy diagrams for the photo-induced charge transit and transfer.

Changes to the manuscript: Fig. R1, R2, and their corresponding description have been added in the Supporting Information as **Supplementary Fig. 18** and **19**, respectively. The following sentence has been added to the paragraph on Page 14: “A further careful study of reduction degree by the electron energy-loss spectroscopy shows that the def-TiO₂ nanorod possesses a ~21 nm-thick reduction shell and the concentration of oxygen vacancies is gradually increased from the inside to the surface (**Supplementary Fig. 18**), corresponding to a gradually increased O distortion. Consequently, the defective amorphous layer and gradually increased O distortion would lead to obviously elevated valence band tail states and gradually elevated valence band maximums in the reduction shell (**Supplementary Fig. 19**), which are much higher than those determined by the absorption spectra. These gradually elevated valence band maximums and valence band tail states can not only promote charge carrier separation but also greatly reduce the energy of holes, suppressing the C-C bonds cleavage. Therefore, with the increase of reduction time, the Y_{CO₂S} over the defective photoanodes are greatly decreased, and no detectable CO₂ is observed after 2s reduction (Entry 4-11 **Table 1**).”

Comment 2: The selective oxidation of glucose has been reported by many approaches, such as electrolysis with high current density and high yield of glucaric acid. What is the advantages of this approach?

Response: The productions of GLA are currently obtained either by microbial fermentation or chemical oxidation and can also be obtained by electrochemical or thermal oxidation.^{4,5} Among these methods, electrochemical oxidation is regarded as a promising approach for the selective oxidation of glucose because the reaction can be operated under mild conditions with the elimination of the use of high-pressure O₂ or hazardous chemical oxidants.⁶ However, electrochemical oxidation needs a large applied potential to drive the reaction, which is energy consuming. In our work, solar photoelectrochemical (PEC) oxidation is applied for the selective oxidation of glucose based on the defective TiO₂ decorated with single-atom Pt, which also operates under mild conditions without high-pressure O₂ or hazardous chemical oxidants. Importantly, this PEC strategy can directly convert solar light into chemical energy, realizing the selective oxidation of glucose at a very low applied potential ($\sim 0.6 V_{\text{RHE}}$). Besides, the single Pt atoms with well-defined active centers greatly reduce the amount of noble metals, thus significantly reducing the catalyst cost. Therefore, the approach via PEC strategy in our work is an easy-to-operate, energy-saving, green, and low-cost approach.

Changes to the manuscript: The following sentence has been added to the paragraph on Page 3: “Besides, PEC cell is the way to drive reactions through the photo-induced electrons and holes directly from the sunlight **with a very low applied potential, which shows good potential for energy saving and environmental protection.**”

Comment 3: Is there any beside products, such as hydrogen?

Response: Yes, the hydrogen is formed on the counter electrode (Pt electrode). Besides, the amount of hydrogen from the Pt/def-TiO₂ photoanode was quantified in **Fig. R3**. The faradaic efficiencies for H₂ over the Pt/def-TiO₂ photoanode have been added based on the J-t plot at 0.6 V_{RHE}. As shown in **Fig R3**, the faradaic efficiencies for H₂ evolution

during 5.5 h PEC glucose oxidation are > 99 %, indicating no side reaction occurred on the counter electrode.

Fig. R3 The J-t plot of the Pt/def-TiO₂ photoanode at 0.6 V_{RHE} measured in 1 M KOH with 10 mM glucose under AM 1.5 G, 100 mW cm⁻² illumination, and the corresponding H₂ evolution.

Changes to the manuscript: Fig. R3 has been added in the Supporting Information as **Supplementary Fig. 12**. The following sentences have been added to the paragraph on Page 13: “The corresponding H₂ evolution on the counter electrode was also examined by online gas chromatography with Ar as carrier gas. As shown in **Supplementary Fig. 12**, the faradaic efficiencies for H₂ evolution during 5.5 h PEC glucose oxidation are > 99%, indicating no side reaction occurred on the counter electrode. The faradaic efficiencies for GLA and GLU are 86.8 and 3.1%, respectively.”

Comment 4: The produced glucaric acid will react with the KOH electrolyte, therefore the real product is corresponding salt. Can it be separated?

Response: The glucaric acid indeed will react with the KOH electrolyte to form the K⁺ glucarate salt. The free-glucaric acid can be obtained from the K⁺ glucarate salt through the method of Amberlyst-15 (H⁺) ion exchange resin based on the previous report ⁷ After the 5.5 h PEC glucose oxidation from the Pt/def-TiO₂ photoanode, the 20 ml

electrolyte was first diluted ten times, and then the obtained solution was mixed with 5 g Amberlyst-15 (H⁺) ion exchange resin. After 10 mins of ion exchange resin, the solution was conducted for the ICP-MS analysis. As shown in **Fig. R4**, 10 min ion exchange resin leads to only 0.83 ppm K⁺ residues in the solution, meaning 99.9% exchange of K⁺ for H⁺. Consequently, the K⁺ glucarate salt can be separated through this ion exchange resin method. Besides, as described in our Method section in SI, all the products are analyzed in a strong acid environment to form their corresponding acids.

Metals analysis on Inductively Coupled Plasma (ICP) – Min sample req: 5mg for solids/5ml for filtered liqs
Standards used in calibration are certified standards prepared in acid/water matrix only.
Digestion: H₂O HNO₃ HCl HF H₂SO₄ AquaRegia hot plate microwave:
 10ml 25ml clear solution ppt observed & filtered

Element	Expected Values (ppm / %)	Determined Values (ppm) (%)		Official Use
K	< 10	0.83		

Fig. R4 The screenshot of ICP results for the solutions with 10 min ion exchange resin, which was obtained from 5.5 h PEC glucose oxidation on the Pt/def-TiO₂ photoanode.

Reviewer 2:

Comment 1: The authors reported an 84.3 % yield of glucaric acid, what is the faradaic efficiency (current efficiency) to GLA? Is there be some other gaseous products, such as O₂, CO₂, CO, etc? Is there H₂O₂ in the liquid product? The authors should report faradaic efficiency for all products, and the calculation method can ref to Adv. Mater. 34, 2201594 (2022).

Response: We thank the reviewer for the constructive comments. According to the suggested method,⁸ the faradaic efficiencies for the products of GLA, GLU, and H₂ over the Pt/def-TiO₂ photoanode have been provided based on the J-t plot at 0.6 V_{RHE}. As shown in **Fig R3**, the faradaic efficiencies for H₂ evolution during 5.5 h PEC glucose oxidation are > 99 %, indicating no side reaction occurred on the counter electrode. Furthermore, the faradaic efficiencies for GLA and GLU are 86.8 and 3.1%, respectively. As shown in **Table R1**. The gaseous products were detected by online gas chromatography (GC) with Ar as carrier gas. During the process of PEC glucose oxidation, no detectable O₂ and CO were observed over all the photoanodes, but some CO₂ was detected over the TiO₂, Pt/TiO₂, and def-TiO₂-2s photoanodes, which are summarized in **Table R1**. The H₂O₂ in the liquid product was examined by using the method from our previous report and no H₂O₂ was detected for all the photoanodes.⁹

Fig. R3 The J-t plot of the Pt/def-TiO₂ photoanode at 0.6 V_{RHE} measured in 1 M KOH with 10 mM glucose under AM 1.5 G, 100 mW cm⁻² illumination, and the corresponding H₂ evolution.

Table R1 Photoelectrochemical oxidation of glucose at 0.6 V_{RHE} in 1 M KOH with glucose.

Entry	photoanode	C _{glucose} (mM)	J (mA cm ⁻²)	t (h)	X _{glucose} (%)	Y _{CO2} (%)	Y _{GLU} (%)	Y _{GLA} (%)	Y _{GLU+GLA} (%)
1	TiO ₂	10	0.41	36	>99.9	24.2	29.7	-	29.7
2	Pt/TiO ₂	10	0.46	36	>99.9	18.4	11.2	26.4	37.6
3	def-TiO ₂ -2s	10	0.78	20	>99.9	7.2	42.4	8.1	50.5
4	def-TiO ₂ -5s	10	1.42	10	98.9	-	69.8	10.2	81.0
5	def-TiO ₂	10	1.86	5.5	97.2	-	77.7	13.8	91.5
6	def-TiO ₂ - 15s	10	1.37	10	72.5	-	68.0	3.1	71.1
7	Pt/def-TiO ₂	10	1.91	5.5	98.8	-	9.2	84.3	93.5
8	Pt/def-TiO ₂	50	2.09	24	91.4	-	13.2	71.4	84.6
9	Pt/def-TiO ₂	100	2.31	48	87.6	-	14.4	63.5	77.9
10	Pt/def-TiO ₂ - 50 cycles	10	1.74	6	97.2	-	47.5	48.4	95.9
11	Pt/def-TiO ₂ - 100 cycles	10	1.41	8	98.3	-	57.1	32.9	95.0
12	Pd/def-TiO ₂	10	1.93	5.5	99.1	-	27.4	65.3	92.7
13	Au/def-TiO ₂	10	1.89	5.5	99.2	-	21.5	69.9	91.4

Changes to the manuscript: We have revised **Table 1** in the manuscript. **Fig. R3** has been added in the Supporting Information as **Supplementary Fig. 12** and the following sentences have been added to the paragraphs on Pages 13 and 14, respectively: “**The corresponding H₂ evolution on the counter electrode was also examined by online gas chromatography with Ar as carrier gas. As shown in Supplementary Fig. 12, the faradaic efficiencies for H₂ evolution during 5.5 h PEC glucose oxidation are > 99%, indicating no side reaction occurred on the counter electrode. The faradaic efficiencies for GLA and GLU are 86.8 and 3.1% respectively.**”; “**The productions of GLU and CO₂ with Y_{GLU} and Y_{CO₂} of ~29.7 and 24.2 %, respectively, were observed by using the TiO₂ photoanode and the no GLA was detected. (Entry 1, Table 1).**”; “**Besides, the Y_{CO₂} on the Pt/TiO₂ photoanode is slightly decreased, probably due to that the promotion of Pt SAs to GLA leads to fewer holes involved in the cleavage of C-C bonds.**”

Comment 2: The authors believe that the TiO₂ NRAs established a low yield of GLA is due to the energy of photo-induced holes is too high for glucose oxidation, leading to cleavage of C-C bonds by most holes. The authors should provide product distributions to verify this claim.

Response: Thank the reviewer’s helpful suggestion. The product distributions yielded by the TiO₂ photoanode have been provided in **Fig. R5**. The product distributions of glucose oxidation over the TiO₂ photoanode from Entry 1, **Table R1** demonstrate that various C₁-C₆ products were produced with only ~30 % C₆ products, suggesting the energy of photo-induced holes is too high for glucose oxidation, leading to cleavage of C-C bonds by most holes.

Fig. R5 Product distribution of glucose on the TiO₂ photoanode from entry 1 in Table 1. Other acids: oxalic, acetic, and formic acids.

Changes to the manuscript: **Fig. R5** has been added in the Supporting Information as **Supplementary Fig. 15**. The following sentences have been added to the paragraph on Page 14: “The product distributions of glucose oxidation over the TiO₂ photoanode from Entry 1, **Table 1** demonstrate that various C₁-C₆ products were produced with only ~30 % C₆ products (**Supplementary Fig. 15**), further confirming the role of high-energy holes in the cleavage of C-C bonds.”

Comment 3: The authors claimed that the Pt/def-TiO₂ photoanode can maintain its activity during its 5 runs of recycled (27.5 h). Long-term photocurrent or 5 runs of recycled photocurrent data is necessary to provide for a more rigorous description.

Response: We thank the reviewer for the professional suggestion and the 5 runs of recycled photocurrents have been provided in **Fig. R6**. The J-t plot of the Pt/def-TiO₂ photoanode in the 1 first run shows that a stable photocurrent can be maintained for the first 3 h, while by prolonging the test to 5.5 h, a significant photocurrent attenuation was observed in the J-t plot. This decreased photocurrent is mainly due to the depletion of glucose and GLU, which is verified by the recovered and reproduced photocurrents

in the 2nd, 3rd, 4th, and 5th runs with the electrolyte refreshed every 5.5 h. These results suggest the good stability of the Pt/def-TiO₂ photoanode.

Fig. R6 Reproducibility of J-t plots of the Pt/def-TiO₂ photoanode at 0.6 V_{RHE} from 5 runs measured in 1 M KOH with 10 mM glucose under AM 1.5 G, 100 mW cm⁻² illumination. The electrolyte was refreshed every 5.5 h.

Changes to the manuscript: Fig. R6 has been added in the Supporting Information as **Supplementary Fig. 13**, and the following sentences have been added to the paragraph on Page 13: “Furthermore, the stability of the Pt/def-TiO₂ photoanode was examined by 5 runs of recycled J-t test (27.5 h) of PEC glucose oxidation. As shown in **Supplementary Fig. 13**, the J-t plot of the Pt/def-TiO₂ photoanode in the 1 first run shows that a stable photocurrent can be maintained for the first 3 h, while by prolonging the test to 5.5 h, a significant photocurrent attenuation was observed. This decreased photocurrent is mainly due to the depletion of glucose and GLU, which is verified by the recovered and reproduced photocurrents in the 2nd, 3rd, 4th, and 5th runs with the electrolyte refreshed every 5.5 h. Besides, after the 27.5 h test, no noticeable J_g recession was observed, as presented in **Supplementary Fig. 14a and b**. These results suggest the good stability of the Pt/def-TiO₂ photoanode.”

Comment 4: As far as I know, in the photocatalytic glucose oxidation reaction, some photocatalysts with less positive valence bands can also break the C-C bonds of glucose to form low-carbon chemicals such as lactic acid (eg. ACS Catal. 12, 11206–11215 (2022); CCS Chem. doi: 10.31635/ccschem.022.202202213 (2022)). Therefore, why Pt/def-TiO₂ can selectively oxidize glucose to GLA but not other C1-C5 chemicals?

Response: Recent reduction engineering on metal oxides, such as black titanium dioxide (TiO_{2-x}), black niobium oxide (Nb₂O_{5-x}), black tungstic oxide (WO_{3-x}), black bismuth vanadate (BiVO_{4-x}) and reduced strontium titanate (SrTiO_{3-x}), can lead to their color changes due to the induced absorption tails by the oxygen vacancies.¹⁰⁻¹⁶ Furthermore, elevated valence band maximum and valence band tail states were also observed in these reduced metal oxides. The valence band tail states accounting for the absorption tails are due to the surface defect derived gap states,^{14,16-18} while the upshift of valence band maximum stems from the distortion of lattice O.³ Previous studies have revealed that the reduction degree decreases from surface to the interior, corresponding to a gradual reduction in the degree of O distortion, thus resulting in gradually elevated valence band maximum in the reduction shell.^{16,17} As shown in **Supplementary Fig. 17**, obvious absorption tails are also observed in our defective TiO₂ photoanodes, meaning the possibility for existence of the gradually elevated valence band maximums. To confirm that, electron energy-loss spectroscopy (EELS) was conducted on the def-TiO₂ electrode, by probing the titanium L_{2,3} edge.^{17,19} As shown in **Fig. R1**, the Ti-L_{2,3} spectrum taken at 1 nm (O₁) from the surface of the def-TiO₂ nanorod shows two main L₃ and L₂ peaks with a separation of 5.3 eV and other three shoulder peaks, indicating that the reduction degree in the surface disorder shell is between Ti₄O₇ and TiO₂.^{17,20-23} While the Ti-L_{2,3} spectrum taken at 28 nm (O₈) from the surface shows the typical EELS spectra of rutile TiO₂. The gradual shift of the Ti-L_{2,3} edge towards higher energies from the surface to the inside (from O₁ to O₆) is related to a gradient descent of the Ti³⁺ content. As a result, the def-TiO₂ electrode shows a ~21 nm-thick reduction shell and the reduction degree gradually increases from the inside to the surface. Therefore,

gradually elevated valance band maximums in the reduction shell and elevated valance band tail states in the surface amorphous layer should be coexisted in the def-TiO₂ electrode(**Fig. R2**), which are much higher than those determined by the absorption spectra. These gradually elevated valance band maximums and valance band tail states not only can promote charge carrier separation but also greatly reduce the energy of holes, suppressing the C-C bonds cleavage.

Supplementary Fig. 17: (a) UV-vis absorption spectra and (b) corresponding tauc plots of defective TiO₂ photoanodes with different reduction times. The bandgaps of defective TiO₂ photoanodes decrease over the reduction time, and up-shift VB maximums should account for the decreased bandgaps.

Fig. R1 (a) STEM image of a def-TiO₂ nanorod with the probing path shown by the red circles and (b) the corresponding EELS spectra of titanium L_{2,3} edge. The pitch between the two neighboring spectra is ~ 3.5 nm.

Fig. R2 Energy band structure of the def-TiO₂ photoanode and corresponding Energy diagrams for the photo-induced charge transit and transfer.

Changes to the manuscript: Fig. R1, R2, and their corresponding description have been added in the Supporting Information as **Supplementary Fig. 18** and **19**, respectively. The following sentences have been added to the paragraph on Page 14: “A further careful study of reduction degree by the electron energy-loss spectroscopy shows that the def-TiO₂ nanorod possesses a ~21nm-thick reduction shell and the content of oxygen vacancies are gradient increased from the inside to the surface (**Supplementary Fig. 18**), corresponding to a gradually increased O distortion. Consequently, the defective amorphous layer and gradually increased O distortion would lead to obviously elevated valence band tail states and gradually elevated valence band maximums in the reduction shell (**Supplementary Fig. 19**), which are much higher than those determined by the absorption spectra. These gradually elevated valence band maximums and valence band tail states not only can promote charge carrier separation but also greatly reduce the energy of holes, suppressing the C-C bonds cleavage. Therefore, with the increase of reduction time, the Y_{CO₂}s over the defective photoanodes are greatly decreased, and no detectable CO₂ is observed after

2s reduction (Entry 4-11 **Table 1**). Benefiting from this, the defective TiO₂ photoanode with 10 s reduction (def-TiO₂) shows a high photocurrent and a high Y_{GLU+GLA} of 91.5% (Entry 5, **Table 1**), however, its Y_{GLA} is still low (13.8 %).”

Comment 5: The authors proved that Pt SAs on def-TiO₂ can significantly enhance the k_t for GLU. In fact, Pt can not only promote the oxidation of alcohol hydroxyl groups, but also promote the oxidation of aldehyde groups according to previous reports (eg. *Angew. Chem. Int. Ed.* **60**, 22908–22914 (2021)). Therefore, the promotion of the oxidation of glucose to GLA by Pt SAs may have an all-round effect.

Response: We agree with the reviewer’s comments that Pt SAs have an all-around effect on the promotion of the oxidation of glucose to GLA. As shown in **Fig. 5b** and **5c**, the k_t s of the Pt/def-TiO₂ photoanode for the oxidation of glucose and GLU are enhanced to 105 and 90 s⁻¹ from these of 95 and 40 s⁻¹, respectively. This suggests that the Pt SAs have an all-around effect on the promotion of aldehyde groups from glucose and alcohol hydroxyl groups from GLA. Importantly, the much lower k_t for GLU is significantly increased to around 90 s⁻¹, which can greatly accelerate the GLA conversion. This reveals that the selectivity of glucose to GLA over the Pt/def-TiO₂ photoanode mainly stems from the accelerated kinetics of the GLU oxidation by the Pt SAs.

Fig. 5 The kinetics of PEC oxidation of glucose (a) ^{13}C - ^1H HMBC-NMR spectrum of the products after 3 h oxidation over the Pt/def-TiO₂ photoanode under AM 1.5 simulated sunlight irradiation. Conditions: 1 M KOH deuteroxide solution with 10 mM glucose, at 0.6 V_{RHE}. The k_t for (b) GLU, (c) GLA, and (d) GUR oxidation over the def-TiO₂, Pt/def-TiO₂, and Pt particles/def-TiO₂ photoanodes measured in 1M KOH aqueous solution 10 mM glucose. (e) Energy diagram for the photo-induced charge transfer and transport based on the Pt/def-TiO₂ photoanode for PEC glucose oxidation.

Changes to the manuscript: The following sentences have been added to the paragraph on Page 19: **The Pt SAs has an all-around effect on the promotion of the oxidation of aldehyde groups from glucose and alcohol hydroxyl groups from GLA, evidenced by their enhanced k_t s on the Pt/def-TiO₂ photoanode (Fig. 5b and 5c). Importantly, the much lower k_t for GLU is significantly increased to around 90 s⁻¹, which can greatly accelerate the GLA conversion.”**

Comment 6: Is Pt a unique metal to promote the selective oxidation of glucose to

GLA? What is the real catalytic site of glucose oxidation?

Response: Pd and Au SAs were also deposited on the def-TiO₂ photoanode to investigate their PEC properties for glucose oxidation. In detail, the def-TiO₂ NRAs were immersed into 1 mmol/L HAuCl₄ and H₂PdCl₄ solutions for 30 mins, respectively. Subsequently, the NRAs on the FTO substrates were annealed at 200 °C for 0.5 h in air to obtain the Pd/def-TiO₂ and Au/def-TiO₂ photoanodes. As shown in **Fig R7**, the TEM, HAADF-STEM, and EDS results suggest that the Pd and Au SAs are successfully deposited on the def-TiO₂ photoanode. Their PEC performances for glucose oxidation were measured at 0.6 V_{RHE} in 1 M KOH with 10 mM glucose under AM 1.5 G, 100 mW cm⁻² illumination. As shown in Entry 12 and 13, **Table R1**, the Pd/def-TiO₂ and Au/def-TiO₂ photoanodes also show significantly enhanced Y_{GLA} of 65.3 and 69.9%, respectively, compared with the def-TiO₂ photoanodes, implying the facilitations of Au and Pd SAs on the conversion of glucose to GLA. However, the Y_{GLAS} from the Pd/def-TiO₂ and Au/def-TiO₂ photoanodes are much lower than that from the Pt/def-TiO₂ photoanode. Therefore, although the Pt SA is not the unique metal to promote the selective oxidation of glucose to GLA, it is the most active single atom for the selective GLA generation.

The real catalytic sites for GLA generation are the Pt SAs. As shown in **Fig. R8b** and **c**, the def-TiO₂ photoanode and Pt/def-TiO₂ photoanodes show similar properties for glucose oxidation but huge differences for GLU oxidation, consistent with the observed kinetic results in **Fig. 5**. It can be concluded that the oxidation of glucose to GLU occurs on the exposed Ti sites or the Pt SAs, while only the Pt SAs sites can further promote the conversion of GLU to GLA.

Fig. R7 (a) TEM, (b) HAADF-STEM images, and (c) the corresponding EDS spectrum of Au/def-TiO₂. (e) TEM, (f) HAADF-STEM images, and (g) the corresponding EDS spectrum of Pd/def-TiO₂.

Table R1 Photoelectrochemical oxidation of glucose at 0.6 V_{RHE} in 1 M KOH with glucose.

Entry	photoanode	C _{glucose} (mM)	J (mA cm ⁻²)	t (h)	X _{glucose} (%)	Y _{CO2} (%)	Y _{GLU} (%)	Y _{GLA} (%)	Y _{GLU+GLA} (%)
1	TiO ₂	10	0.41	36	>99.9	24.2	29.7	-	29.7
2	Pt/TiO ₂	10	0.46	36	>99.9	18.4	11.2	26.4	37.6
3	def-TiO ₂ -2s	10	0.78	20	>99.9	7.2	42.4	8.1	50.5
4	def-TiO ₂ -5s	10	1.42	10	98.9	-	69.8	10.2	81.0
5	def-TiO ₂	10	1.86	5.5	97.2	-	77.7	13.8	91.5
6	def-TiO ₂ - 15s	10	1.37	10	72.5	-	68.0	3.1	71.1
7	Pt/def-TiO ₂	10	1.91	5.5	98.8	-	9.2	84.3	93.5

8	Pt/def-TiO ₂	50	2.09	24	91.4	-	13.2	71.4	84.6
9	Pt/def-TiO ₂	100	2.31	48	87.6	-	14.4	63.5	77.9
10	Pt/def-TiO ₂ - 50 cycles	10	1.74	6	97.2	-	47.5	48.4	95.9
11	Pt/def-TiO ₂ - 100 cycles	10	1.41	8	98.3	-	57.1	32.9	95.0
12	Pd/def-TiO ₂	10	1.93	5.5	99.1	-	27.4	65.3	92.7
13	Au/def-TiO ₂	10	1.89	5.5	99.2	-	21.5	69.9	91.4

Fig. R8 LSV profiles of (a) the TiO₂, def-TiO₂, and Pt/def-TiO₂ photoanodes for GLU, GUR,

and GLA oxidation measured in 1 M KOH with 10 mM GLU, GUR, and GLA, respectively, under AM 1.5 G, 100 mW cm⁻² illumination and dark.

Changes to the manuscript: Fig. R7, R8, and their corresponding descriptions have been added in the Supporting Information as **Supplementary Fig. 23** and **30**, respectively. **Table 1** has been updated by **Table R1** in the manuscript. The following sentences have been added to the paragraphs on Pages 15 and 19, respectively: “**Besides, the Au and Pd SAs were also successfully deposited on the def-TiO₂ (Au/def-TiO₂ and Pd/def-TiO₂) photoanodes through an immersion method to investigate their PEC performances for glucose oxidation (Supplementary Fig. 23). As shown in Entry 12 and 13, Table 1, the Pd/def-TiO₂ and Au/def-TiO₂ photoanodes also show significantly enhanced Y_{GLA} of 65.3 and 69.9%, respectively, compared with the def-TiO₂ photoanodes, implying the facilitations of Au and Pd SAs on the conversion of glucose to GLA. However, the Y_{GLAS} from the Pd/def-TiO₂ and Au/def-TiO₂ photoanodes are much lower than that from the Pt/def-TiO₂ photoanode. Therefore, although the Pt SA is not the unique metal to promote the selective oxidation of glucose to GLA, it is the most active single atom for the selective GLA generation.**”; “**Besides, LSV curves for GLU, GUR, and GLA oxidation over the TiO₂, def-TiO₂, and Pt/def-TiO₂ photoanodes were tested with and without illumination (Supplementary Fig. 30), which further confirmed the PEC reaction for glucose oxidation and fast kinetics for GUR oxidation. These LSV results also reveal the accelerated GLU oxidation over the Pt SAs and sluggish GLA oxidation, consequently resulting in a high selectivity of glucose to GLA over the Pt/def-TiO₂ photoanode.**”

Comment 7: The authors provide proposed possible reaction pathway from glucose to GLA, but it is slightly simple. And how H₂O is involved in the oxidation of glucose, and the charge transfer at each step have not been elucidated. Can the authors obtain some more accurate intermediates their evolution and transformation process through some in situ characterization, such as ESR and DRIFTS.

Response: To elucidate the reaction pathway of glucose oxidation, electron paramagnetic resonance (EPR) spectroscopy and *in-situ* Fourier transform infrared spectra (FTIR) measurements were carried out to monitor the reaction intermediates. **Fig. R9a** shows the *in-situ* FTIR of the Pt/def-TiO₂ photoanode in the reaction of PEC glucose oxidation recorded at 0.6 V_{RHE} from 0 to 180 min. The negative characteristic IR bands located at ~1350, 1415, 1585, 1640, and 1740 cm⁻¹ are assigned to $\delta(\text{CH}_2)$ in the C6-position of gluconate anion and glucose, symmetric $\nu_s(\text{O}-\text{C}-\text{O})$ of COO⁻ function in gluconate (C1-position), L-gluconate (C1-position) and glucarate (C1-position and C6-position), asymmetric stretching vibration $\nu_{\text{as}}(\text{O}-\text{C}-\text{O})$ of COO⁻ function, $\delta(\text{H}_2\text{O})$ due to interfacial water absorbed on the photoanode, and $\nu(\text{C}=\text{O})$ in gluconolactone or L-gluconate (C6-position), respectively.²⁴⁻²⁹ The $\nu_s(\text{O}-\text{C}-\text{O})$ and $\nu_{\text{as}}(\text{O}-\text{C}-\text{O})$ IR bands show obvious dependence on PEC duration, suggesting the continuous generation of the COO⁻ function. Remarkably, no obvious C-C breaking occurred during the PEC glucose oxidation over the Pt/def-TiO₂ photoanode as revealed by the undetected C-C bond cleavage compounds (CO: 1900–2100 cm⁻¹, carbonate: 1396 cm⁻¹, CO₂: 2340–2350 cm⁻¹).²⁴ The stable $\delta(\text{H}_2\text{O})$ IR bands during the PEC glucose oxidation indicate the involvement of water in the reaction. Further EPR spectrum exhibits quartet signals with an intensity ratio of 1:2:2:1, assigned to the DMPO-•OH adduct (**Fig. R9b**).³⁰ This means that the absorbed water molecules on the catalyst are oxidated by the holes to form •OH radicals. Meanwhile, •OH radicals and holes will separately attack the H atom of the C1-H, and the C1-O-H bonds, resulting in the formation of C₁-OO⁻ groups.^{24,25} Besides, the peak intensities of $\delta(\text{CH}_2)$ remain almost unchanged until 90 min and then decrease gradually with increased $\nu(\text{C}=\text{O})$ IR bands, suggesting that the oxidation process mainly occurred on the C6 position after 90 min. These results suggest that the transformation of glucose into GLA is achieved through the intermediate production of GLU and GUR.

A reaction mechanism for the PEC oxidation of glucose to GLA on the Pt/def-TiO₂ photoanode can therefore be proposed (**Fig R9c**). The holes from the photoanode will react with the absorbed water to form absorbed •OH radicals,^{28,31} which can abstract the H atoms of the C1-H and C1-O-H bonds from absorbed C1 of glucose on the

photoanode forming C1=O bond in step 2.^{25,32} Subsequently, the C5-O bond is split by hydrolysis, and then the GLU is formed through desorption from the photoanode.^{24,25} Previous study has shown that highly dispersed Pt catalysts are capable to selectively activate primary alcohols for further oxidation.³³ Therefore, our Pt SAs on the photoanode are proposed to selectively adsorb primary alcohols from the C6 position. After the co-adsorption of C6-OH on the Pt SAs and C6 on the photoanode (step 4), the absorbed •OH radicals will react with the H atoms of C6-OH and C6-H, separately, to form a C6=O bond (step 5), where some products desorb from the catalyst to generate GUR and others will adjust their adsorption states for the subsequent reactions. During step 6, with the holes and hydroxyl groups, the C6=O bond will be activated and a new C6-OH bond will be formed.³⁴ Subsequently, the groups from the C6 position will follow the same reaction pathway as the C1 position to yield GLA, eventually. Therefore, both the Pt SAs and absorbed •OH radicals can play key roles in the selective PEC oxidation of glucose to GLA.

Fig. R9 (a) *In-situ* Fourier transform infrared spectra for PEC glucose conversion on the Pt/def-TiO₂ photoanode in 1 M KOH with 10 mM glucose. (b) EPR detection of •OH using DMPO as a spin-trapping agent under illumination with Pt/def-TiO₂ in 1 M KOH. (c) Schematic illustration of the possible pathway for the PEC oxidation of glucose to GLU and GLA over the Pt/def-TiO₂ photoanode.

Changes to the manuscript: Fig. R9 has been added in the manuscript as Fig. 6. The following sentences have been added to the paragraphs on Page 22: “To elucidate the reaction pathway of glucose oxidation, electron paramagnetic resonance (EPR)

spectroscopy, and *in-situ* Fourier transform infrared spectra (FTIR) measurements were carried out to monitor the reaction intermediates. **Fig. 6a** shows the *in-situ* FTIR of the Pt/def-TiO₂ photoanode in the reaction of PEC glucose oxidation recorded at 0.6 V_{RHE} from 0 to 180 min. The negative characteristic IR bands located at ~1350, 1415, 1585, 1640, and 1740 cm⁻¹ are assigned to $\delta(\text{CH}_2)$ in the C6-position of gluconate anion and glucose, symmetric $\nu_s(\text{O}-\text{C}-\text{O})$ of COO⁻ function in gluconate (C1-position), L-glucuronate (C1-position) and glucarate (C1-position and C6-position), asymmetric stretching vibration $\nu_{as}(\text{O}-\text{C}-\text{O})$ of COO⁻ function, $\delta(\text{H}_2\text{O})$ due to interfacial water absorbed on the photoanode, and $\nu(\text{C}=\text{O})$ in gluconolactone or L-glucuronate (C6-position), respectively.²⁴⁻²⁹ The $\nu_s(\text{O}-\text{C}-\text{O})$ and $\nu_{as}(\text{O}-\text{C}-\text{O})$ IR bands show obvious dependence on PEC duration, suggesting the continuous generation of the COO⁻ function. Remarkably, no obvious C-C breaking occurred during the PEC glucose oxidation over the Pt/def-TiO₂ photoanode as revealed by the undetected C-C bond cleavage compounds (CO: 1900–2100 cm⁻¹, carbonate: 1396 cm⁻¹, CO₂: 2340–2350 cm⁻¹).²⁴ The stable $\delta(\text{H}_2\text{O})$ IR bands during the PEC glucose oxidation indicate the involvement of water in the reaction. Further EPR spectrum exhibits quartet signals with an intensity ratio of 1:2:2:1, assigned to the DMPO-•OH adduct (**Fig. 6b**).³⁰ This means that the absorbed water molecules on the catalyst are oxidated by the holes to form •OH radicals. Meanwhile, •OH radicals and holes will separately attack the H atom of the C1-H, and the C1-O-H bonds, resulting in the formation of C₁-OO• groups.^{24,25} Besides, the peak intensities of $\delta(\text{CH}_2)$ remain almost unchanged until 90 min and then decrease gradually with increased $\nu(\text{C}=\text{O})$ IR bands, suggesting that the oxidation process mainly occurred on the C6 position after 90 min. These results suggest that the transformation of glucose into GLA is achieved through the intermediate production of GLU and GUR.

A reaction mechanism for the PEC oxidation of glucose to GLA on the Pt/def-TiO₂ photoanode can therefore be proposed (**Fig 6c**). The holes from the photoanode will react with the absorbed water to form absorbed •OH radicals,^{28,31} which can abstract the H atoms of the C1-H and C1-O-H bonds from absorbed C1 of glucose on the photoanode forming C1=O bond in step 2.^{25,32} Subsequently, the C5-O bond is split by

hydrolysis, and then the GLU is formed through desorption from the photoanode.^{24,25} Previous study has shown that highly dispersed Pt catalysts are capable to selectively activate primary alcohols for further oxidation.³³ Therefore, our Pt SAs on the photoanode are proposed to selectively adsorb primary alcohols from the C6 position. After the co-adsorption of C6-OH on the Pt SAs and C6 on the photoanode (step 4), the absorbed •OH radicals will react with the H atoms of C6-OH and C6-H, separately, to form a C6=O bond (step 5), where some products desorb from the catalyst to generate GUR and others will adjust their adsorption states for the subsequent reaction. During step 6, with the holes and hydroxyl groups, the C6=O bond will be activated and a new C6-OH bond will be formed.³⁴ Subsequently, the groups from the C6 position will follow the same reaction pathway as the C1 position to yield GLA, eventually. Therefore, both the Pt SAs and absorbed •OH radicals can play key roles in the selective PEC oxidation of glucose to GLA.”

Comment 8: Other minor revisions:

- 1) **Statements and abbreviations for nanorod arrays are not uniform in this manuscript, and there are spelling errors such as in lines 82 and 83.**
- 2) **In this manuscript, in line 148, the growth direction of TiO₂ NRAs is described, but Fig. 3a is the growth direction of Pt/def-TiO₂ NRAs.**
- 3) **In the SI, in line 149-150, “When compared to the Pt/def-TiO₂ photoanodes, the Mott-Schottky changes in the Pt/def-TiO₂ photoanodes are not obvious” should be corrected to “When compared to the def-TiO₂ photoanodes, the Mott Schottky changes in the Pt/def-TiO₂ photoanodes are not obvious”.**
- 4) **The Methods section should be described in more detail, e.g. photoanode size, cell volume, electrolyte volume.**

Response: We thank the reviewer for the comments and careful revisions have been made to the manuscript and SI. The Methods section has been tied up in more detail.

Changes to the manuscript: The “Methods” section has been tied up as follows: “The PEC performance of the samples was measured in a three-electrode system with an electrochemical workstation (CHI 760E) under AM 1.5G simulated sunlight of 100 mW cm⁻². The simulated solar illumination was obtained from a 300 W Xenon lamp with an AM 1.5G filter (100 mW cm⁻²). Samples on FTO substrates were used directly as the working electrode, with a Pt wire and an Ag/AgCl (KCl saturated) electrode as counter and reference electrodes respectively. **The active areas for the working and counter electrodes are 1.5×2.1 cm² and 1 cm², respectively.** All the samples were illuminated through the sample side (front-side illumination). The PEC performance was recorded in an H-type PEC cell, in which **30 ml cathode and 30 ml anode chambers** were separated by an anion-exchange membrane (Fumasep FAA-3-PK-130). **The anode chamber was filled with 20 ml, 1 M KOH solution dissolved with 10 mM glucose and the cathode chamber electrolyte was filled with 20 ml, 1 M KOH solution. The reaction temperature for all photoanodes is maintained at 20 °C.** Mott-Schottky plots were derived from impedance-potential tests conducted at a frequency of 1 kHz in dark. Intensity-modulated photocurrent spectroscopy (IMPS), small perturbation transient photocurrent measurements, and incident photon-to-current conversion efficiency (IPCE) were recorded by the Zahner Zennium C-IMPS system. **The IPCE value was measured at 0.6 V_{RHE} under various monochromatic light irradiation and calculated by the following equation:**

$$\text{IPCE} = \frac{J(\text{mA cm}^{-2}) \times 1239.8(\text{V nm})}{\lambda(\text{nm}) \times P_{\text{mono}}(\text{mW cm}^{-2})} \times 100\% \quad (\text{S1})$$

where J is the photocurrent density, λ is the wavelength of the incident light, and P_{mono} is the illumination intensity at different wavelengths. The incident photo-to-GLA conversion efficiency of the Pt/def-TiO₂ photoanode at 0.6 V_{RHE}. Was calculated by the following equation:

$$\text{Incident photo – to – GLA conversion efficiency} = \text{IPCE} \times f_{\text{GLA}} \quad (\text{S2})$$

where f_{GLA} is the faradaic efficiency for GLA production.

Quantification analysis of the reaction products: The glucose and its products from the PEC cell were analyzed by a Shimadzu LC-20AT high-performance liquid

chromatography (HPLC) equipped with a refractive index detector. 5 mM H₂SO₄ at a flow rate of 0.6 mL min⁻¹ was used as the mobile phase. In each analysis, ten times diluted electrolyte withdrawn from the PEC cell was injected directly into a BioRad Aminex 87H column with a column temperature of 60 °C. The glucose and its products were identified and quantification analyzed by comparing their retention times in the chromatograms with those of the standard solution. Gaseous reduction products were analyzed using an online gas chromatograph with Ar as the carrier gas (GC, Shimadzu 2014). . The conversion (X_G), yield (Y_{products}), and selectivity (S_{products}) were calculated using the following equations:

$$X_G = \frac{n_{Gi} - n_{G0}}{n_{G0}} \times 100\% \quad (S3)$$

$$Y_{\text{products}} = \frac{n_{\text{products}}}{n_{G0}} \times 100\% \quad (S4)$$

$$S_{\text{products}} = \frac{n_{\text{products}}}{n_{Gi} - n_{G0}} \times \frac{\mu}{\varphi} \times 100\% \quad (S5)$$

where n_{G0}, n_{Gi}, and n_{products}, are the initial mole number of glucose, the residual number of glucose, and the generated mole number of the products. The μ and φ represent the stoichiometric coefficients of the reaction.

The faradaic efficiencies (f_{products}) for the products were calculated by the following equation:

$$f_{\text{products}} = \frac{n_{\text{products}} \times m \times F}{Q} \times 100\% \quad (S6)$$

where m represents the quantities of charge required for the generation of one product, F is Faraday's constant (96485.33 C mol⁻¹), and Q is the total charge.”

“EPR trapping measurements: 5, 5-dimethyl-1-pyrroline N-oxide (DMPO) was used to trap the generated hydroxyl radical in the reaction system at room temperature. Briefly, 0.5 mg Pt/def-TiO₂ powders collected from the FTO substrates were dispersed in 2 mL of 1 M KOH solution, and then 50 μL DMPO was added into the solution. The mixture was then filled with Ar and sonicated for 1 min. After the sonication, the solution was irradiated by a 300 W xenon lamp with an AM 1.5 filter for 30 s. The resulting solution was subjected to analysis by using a JEOL (FA200) ESR Spectrometer.

***In-situ* Fourier transform infrared spectroscopy (FTIR) measurements:** 5 mg of Pt/def-TiO₂ NRAs detached from the FTO substrates and 80 μL of 5 wt % Nafion solution were dispersed in 2 mL of 4:1 v/v water/ethanol by sonication for 1 h to form a homogeneous ink. 30 μL of ink dispersion was dropped onto the central area (confined by an O-ring with Φ=8 mm) of an Au film chemically deposited on the basal plane of a hemicylindrical Si prism. The Si prism as working electrode was assembled in a spectroelectrochemical cell with Pt wire as counter electrode, Ag/AgCl electrode as reference electrode, and 1 M KOH with 10 mM glucose as electrolytes. In-situ FTIR spectra were measured on a Nicolet iS50 spectrometer, equipped with an MCT cryogenic detector with a resolution of 4 cm⁻¹ each single-beam spectrum was an average of 200 scans. A CHI 760e electrochemistry workstation (Shanghai CH Instruments, Inc.) was used for potential control. A light-emitting diode (LED) ultraviolet light (365 nm) was used as the light source for the PEC experiments.”

Reviewer 3:

Comment 1: The amount of CO₂ under low GLA+GLU yield conditions in glucose oxidation should be added.

Response: As suggested by the reviewer, the amounts of CO₂ over all the photoanodes have been added in **Table R1**.

Table R1 Photoelectrochemical oxidation of glucose at 0.6 V_{RHE} in 1 M KOH with glucose.

Entry	photoanode	C _{glucose} (mM)	J (mA cm ⁻²)	t (h)	X _{glucose} (%)	Y _{CO2} (%)	Y _{GLU} (%)	Y _{GLA} (%)	Y _{GLU+GLA} (%)
1	TiO ₂	10	0.41	36	>99.9	24.2	29.7	-	29.7
2	Pt/TiO ₂	10	0.46	36	>99.9	18.4	11.2	26.4	37.6
3	def-TiO ₂ -2s	10	0.78	20	>99.9	7.2	42.4	8.1	50.5
4	def-TiO ₂ -5s	10	1.42	10	98.9	-	69.8	10.2	81.0
5	def-TiO ₂	10	1.86	5.5	97.2	-	77.7	13.8	91.5
6	def-TiO ₂ - 15s	10	1.37	10	72.5	-	68.0	3.1	71.1
7	Pt/def-TiO ₂	10	1.91	5.5	98.8	-	9.2	84.3	93.5
8	Pt/def-TiO ₂	50	2.09	24	91.4	-	13.2	71.4	84.6
9	Pt/def-TiO ₂	100	2.31	48	87.6	-	14.4	63.5	77.9
10	Pt/def-TiO ₂ - 50 cycles	10	1.74	6	97.2	-	47.5	48.4	95.9

11	Pt/def-TiO ₂ - 100 cycles	10	1.41	8	98.3	-	57.1	32.9	95.0
12	Pd/def-TiO ₂	10	1.93	5.5	99.1	-	27.4	65.3	92.7
13	Au/def-TiO ₂	10	1.89	5.5	99.2	-	21.5	69.9	91.4

Changes to the manuscript: Table 1 has been updated by Table R1 in the manuscript.

The following sentences have been added to the paragraphs on Pages 14 and 15, respectively: “The productions of GLU and CO₂ with Y_{GLU} and Y_{CO₂} of ~29.7 and 24.2 %, respectively, were observed by using the TiO₂ photoanode and the no GLA was detected. (Entry 1, Table 1).”; “Besides, the Y_{CO₂} on the Pt/TiO₂ photoanode is slightly decreased, probably due to that the promotion of Pt SAs to GLA leads to fewer holes involved in the cleavage of C-C bonds.”; “Therefore, with the increase of reduction time, the Y_{CO₂}s over the defective photoanodes are greatly decreased, and no detectable CO₂ is observed after 2s reduction (Entry 4-11 Table 1). Benefiting from this, the defective TiO₂ photoanode with 10 s reduction (def-TiO₂) shows a high photocurrent and a high Y_{GLU+GLA} of 91.5% (Entry 5, Table 1), however, its Y_{GLA} is still low (13.8 %).”

Comment 2: The LSV curves for GLU, GUR, and GLA oxidation should be added using Pt, TiO₂, def-TiO₂, and Pt/def-TiO₂ (photo)electrodes under light and dark conditions, respectively.

Response: As suggested by the reviewer, the LSV curves for GLU, GUR, and GLA oxidation over the TiO₂, def-TiO₂, and Pt/def-TiO₂ photoanodes were tested with and without illumination in the electrolytes of 1 M KOH with 10 mM GLU, GUR, and GLA, respectively. As shown in Fig R8, the dark currents for the GLU, GUR, and GLA oxidations over all the photoanodes can be neglected compared to their corresponding photocurrents, suggesting that the GLU, GUR, and GLA oxidations over all the

photoanodes are PEC reactions. Besides, the photocurrent densities of all the photoanodes follow this sequence: $J_{\text{GUR}} > J_{\text{GLU}} > J_{\text{GLA}}$, further revealing the fast kinetics for GUR oxidation. The TiO_2 photoanode shows much smaller differences in the J_{GUR} , J_{GLU} , and J_{GLA} than the def- TiO_2 , and Pt/def- TiO_2 photoanodes, probably because the cleavage of C-C bonds is the main reaction on the TiO_2 photoanode. With the cleavage of C-C bonds suppressed, the def- TiO_2 photoanode shows distinctly different photocurrents for the GLU, GUR, and GLA oxidations. The J_{GLU} is obviously smaller than J_{glucose} (**Fig. 4a**) and J_{GUR} (**Fig R8b**), also confirming the rate-limiting step of GLU oxidation on the def- TiO_2 photoanode. The further deposition of Pt SAs significantly promotes GLU oxidation (**Fig R8c**), accelerating the conversion of GLA from GLU. Furthermore, sluggish kinetics for GLA oxidation are observed on the Pt/def- TiO_2 photoanode. The accelerated GLU oxidation and sluggish GLA oxidation consequently result in a high selectivity of glucose to GLA over the Pt/def- TiO_2 photoanode.

Fig. R8 LSV profiles of (a) the TiO_2 , def- TiO_2 , and Pt/def- TiO_2 photoanodes for GLU, GUR,

and GLA oxidation measured in 1 M KOH with 10 mM GLU, GUR, and GLA, respectively, under AM 1.5 G, 100 mW cm⁻² illumination and dark.

The glucose oxidation realized in our studies is a PEC method based on the Pt/def-TiO₂ photoanode. Actually, the charge carrier dynamics in the PEC process and electrochemical process are totally different. During the electrochemical process, the electron Fermi level can be regulated by the applied potential, thus the energy of the electron involved reaction is regulated by the applied potential. Consequently, the selectivity in an electrochemical reaction can be modulated by the applied potential. Moggia et. al. has reported that the electrochemical glucose oxidation over the Pt electrode exhibited different selectivity by varying the applied potential.³⁵ However, the charge carrier process for PEC reaction is totally different. For a PEC reaction, a semiconductor is used as the photoelectrode, when the semiconductor contacts with the electrolyte, the charge transfer at the interface will lead to a band bending at the surface of the photoelectrode. Besides, surface states of the photoelectrode will lead to Fermi level pinning. Therefore, the applied potential can only regulate the degree of band bending in the space charge layer instead of the Fermi level.³⁶ For PEC oxidation, the variation of applied potential within a certain range (usually <1.6 V_{RHE}) can only modulate the charge carrier separation and have tiny effects on the energy of the electron and hole involved reaction. As for our PEC glucose oxidation, the energy of the electron and hole involved reaction is modulated by the reduction treatment, and the Pt/def-TiO₂ photoanode also shows almost identical X_{glucose} and Y_{GLA} at different applied potentials (Fig. 4i). Therefore, the supplement of LSV using the Pt electrode, which is an electrochemical reaction, can only provide limited information to our discussion of PEC processes.

Changes to the manuscript: Fig. R8 and its corresponding description have been added in the Supporting Information as **Supplementary Fig. 30**. The following sentences have been added to the paragraph on Page 19: “Besides, LSV curves for GLU, GUR, and GLA oxidation over the TiO₂, def-TiO₂, and Pt/def-TiO₂ photoanodes were tested with and without illumination (**Supplementary Fig. 30**), which further confirms the PEC reaction for glucose oxidation and fast kinetics for GUR oxidation. These LSV

results also reveal the accelerated GLU oxidation over the Pt SAs and sluggish GLA oxidation, consequently resulting in a high selectivity of glucose to GLA over the Pt/def-TiO₂ photoanode.”

Comment 3: In Fig. 5, the authors should add (1) oxidation rates at low applied bias, including no applied bias conditions, (2) faradaic efficiencies of the oxidation products, and (3) the amount of CO₂ production.

Response: We thank the reviewer for the helpful suggestions. Since the oxidation rates were measured by intensity-modulated photocurrent spectroscopy (IMPS), a technology that requires reaction current to flow through the cell (**Supplementary Fig. 10**).³⁷⁻⁴⁰ Under the conditions of no applied bias, no reaction current flow through the cell, therefore, we cannot provide the oxidation rates without applied bias. As shown in **Fig R10**, except the Pt/def-TiO₂ photoanodes, other photoanodes showed no anode photocurrents of the GLU oxidation below 0.3 V_{RHE}, leading to unobtainable oxidation rates below 0.3 V_{RHE} for these photoanodes. Therefore, additional oxidation rates at 0.4 V_{RHE} have been added as shown in **Fig R10** and their corresponding recombination rates have replaced the original versions. The amount of CO₂ production has been provided in **Table R1**. The faradaic efficiencies for the products of GLA, GLU, and H₂ over the Pt/def-TiO₂ photoanode have been provided based on the J-t plot at 0.6 V_{RHE}. As shown in **Fig R1**, the faradaic efficiencies for H₂ evolution during 5.5 h PEC glucose oxidation are > 99 %, indicating no side reaction occurred on the counter electrode. Furthermore, the faradaic efficiencies for GLA and GLU are 86.8 and 3.1%, respectively.

Fig. R10 The kinetics of PEC oxidation of glucose (a) ^{13}C - ^1H HMBC-NMR spectrum of the products after 3 h oxidation over the Pt/def-TiO₂ photoanode under AM 1.5 simulated sunlight irradiation. Conditions: 1 M KOH deuteroxide solution with 10 mM glucose, at 0.6 V_{RHE} . The k_t for (b) GLU, (c) GLA, and (d) GUR oxidation over the def-TiO₂, Pt/def-TiO₂, and Pt particles/def-TiO₂ photoanodes measured in 1M KOH aqueous solution 10 mM glucose. (e) Energy diagram for the photo-induced charge transfer and transport based on the Pt/def-TiO₂ photoanode for PEC glucose oxidation.

Changes to the manuscript: Fig. 5 has been updated by Fig. R10, and k_{rec} and η_{tran} in Supplementary Fig. 27-29 also have been updated. The following sentences have been added to the paragraphs on Pages 13, 14, and 15, respectively: “The corresponding H₂ evolution on the counter electrode was also examined by online gas chromatography with Ar as carrier gas. As shown in Supplementary Fig. 12, the faradaic efficiencies for H₂ evolution during 5.5 h PEC glucose oxidation are > 99%, indicating no side reaction occurred on the counter electrode. The faradaic efficiencies for GLA and GLU are 86.8 and 3.1%, respectively.”; “The productions of GLU and CO₂ with Y_{GLU} and

Y_{CO_2} of ~29.7 and 24.2 %, respectively, were observed by using the TiO_2 photoanode and the no GLA was detected. (Entry 1, **Table 1**).”; “Besides, the Y_{CO_2} on the Pt/ TiO_2 photoanode is slightly decreased, probably due to that the promotion of Pt SAs to GLA leads to fewer holes involved in the cleavage of C-C bonds.”; “Therefore, with the increase of reduction time, the Y_{CO_2} s over the defective photoanodes are greatly decreased, and no detectable CO_2 is observed after 2s reduction (Entry 4-11 **Table 1**). Benefiting from this, the defective TiO_2 photoanode with 10 s reduction (def- TiO_2) shows a high photocurrent and a high $Y_{GLU+GLA}$ of 91.5% (Entry 5, **Table 1**), however, its Y_{GLA} is still low (13.8 %).”

Comment 4: The role of Pt is unclear. If oxidation of GLU to GLA proceeds over Pt, is it a photoelectrochemical reaction? In particular, for the oxidation of intermediates, is it possible that the reaction proceeds photo or thermally without needing an applied voltage? The authors need to discuss these points further.

Response: We thank the reviewer for the helpful suggestions. The reaction temperature during the PEC oxidation for all the photoanodes was maintained at 20 °C through a water bath device. To figure out whether the oxidation of GLU to GLA proceeds over Pt SAs is a photoelectrochemical reaction, the Pt/def- TiO_2 NRAs were separately immersed into the electrolytes of 1 M KOH with 10 mM glucose, 1 M KOH with 10 mM GLU, and 1 M KOH with 10 mM GUR without any applied voltage. After 5 h illumination of simulated sunlight, the electrolytes were analyzed by HPLC and the results show that trace amounts of GLA were detected in the electrolytes, suggesting that the oxidation of GLU to GLA proceeds over Pt SAs is indeed a PEC reaction.

Changes to the manuscript: The Methods section has been tied up in more detail and the following sentences have been added to the paragraph on Page 18: “Furthermore, the Pt/def- TiO_2 NRAs were separately immersed into the electrolytes of 1 M KOH with 10 mM glucose, 1 M KOH with 10 mM GLU, and 1 M KOH with 10 mM GUR without any applied voltage. After 5 h illumination of simulated sunlight at 20 °C, the electrolytes were analyzed by HPLC and the results show that trace amounts of GLA

were detected in the electrolytes, suggesting that the oxidation of GLU to GLA proceeds over Pt SAs is indeed a PEC reaction.”.

Reference

- 1 Tian, Z. *et al.* Hydrogen plasma reduced black TiO₂-B nanowires for enhanced photoelectrochemical water-splitting. *J. Power Sources* **325**, 697-705 (2016).
- 2 Tian, Z. *et al.* Highly Conductive Cable-Like Bicomponent Titania Photoanode Approaching Limitation of Electron and Hole Collection. *Adv Funct Mater* **28** (2018).
- 3 Liu, L., Yu, P. Y., Chen, X., Mao, S. S. & Shen, D. Z. Hydrogenation and disorder in engineered black TiO₂. *Phys. Rev. Lett.* **111**, 065505 (2013).
- 4 Mehlretter, C., Rist, C. J. J. o. A. & Chemistry, F. Sugar oxidation, saccharic and oxalic acids by the nitric acid oxidation of dextrose. *J. Agric. Food Chem.* **1**, 779-783 (1953).
- 5 Mustakas, G., Slotter, R., Zipf, R. J. I. & Chemistry, E. Pilot plant potassium acid saccharate by nitric acid oxidation of dextrose. *Ind. Eng. Chem.* **46**, 427-434 (1954).
- 6 Liu, W. J. *et al.* Efficient electrochemical production of glucaric acid and H₂ via glucose electrolysis. *Nat. Commun.* **11**, 265 (2020).
- 7 Armstrong, R. D., Kariuki, B. M., Knight, D. W. & Hutchings, G. J. How to Synthesise High Purity, Crystalline d-Glucaric Acid Selectively. *Eur. J. Org. Chem.* **2017**, 6811-6814 (2017).
- 8 Wang, Z. *et al.* Boosting H₂ Production from a BiVO₄ Photoelectrochemical Biomass Fuel Cell by the Construction of a Bridge for Charge and Energy Transfer. *Adv. Mater.* **34**, e2201594 (2022).
- 9 Tian, Z. *et al.* Efficient photocatalytic hydrogen peroxide generation coupled with selective benzylamine oxidation over defective ZrS₃ nanobelts. *Nat. Commun.* **12**, 1-10 (2021).
- 10 Wang, G. *et al.* Hydrogen-treated TiO₂ nanowire arrays for photoelectrochemical water splitting. *Nano letters* **11**, 3026-3033 (2011).
- 11 Wang, G. *et al.* Hydrogen-treated WO₃ nanoflakes show enhanced photostability. *Energy Environ. Sci.* **5**, 6180-6187 (2012).

- 12 Cui, H. L. *et al.* Black nanostructured Nb₂O₅ with improved solar absorption and enhanced photoelectrochemical water splitting. *J. Mater. Chem. A* **3**, 11830-11837 (2015).
- 13 Zhao, W. L. *et al.* Black strontium titanate nanocrystals of enhanced solar absorption for photocatalysis. *Crystengcomm* **17**, 7528-7534 (2015).
- 14 Chen, X., Liu, L., Yu, P. Y. & Mao, S. S. Increasing solar absorption for photocatalysis with black hydrogenated titanium dioxide nanocrystals. *Science* **331**, 746-750 (2011).
- 15 Coh, S. *et al.* Alternative structure of TiO₂ with higher energy valence band edge. *Phys. Rev. B* **95** (2017).
- 16 Tian, Z. *et al.* Novel black BiVO₄/TiO_{2-x} photoanode with enhanced photon absorption and charge separation for efficient and stable solar water splitting. *Adv. Energy Mater.* **9**, 1901287 (2019).
- 17 Tian, M. *et al.* Structure and Formation Mechanism of Black TiO₂ Nanoparticles. *Acs Nano* **9**, 10482-10488 (2015).
- 18 Zhang, K. *et al.* Overcoming Charge Collection Limitation at Solid/Liquid Interface by a Controllable Crystal Deficient Overlayer. *Adv. Energy Mater.* **7**, 1600923 (2017).
- 19 Folger, A., Kalb, J., Schmidt-Mende, L. & Scheu, C. Tuning the Electronic Conductivity in Hydrothermally Grown Rutile TiO₂ Nanowires: Effect of Heat Treatment in Different Environments. *Nanomaterials (Basel)* **7** (2017).
- 20 Bertoni, G. *et al.* Quantification of crystalline and amorphous content in porous samples from electron energy loss spectroscopy. *Ultramicroscopy* **106**, 630-635 (2006).
- 21 Gao, Q. *et al.* Direct Evidence of Lithium-Induced Atomic Ordering in Amorphous TiO₂ Nanotubes. *Chem. Mater.* **26**, 1660-1669 (2014).
- 22 Azor-Lafarga, A. *et al.* Modified Synthesis Strategies for the Stabilization of low n Ti_n O_{2n-1} Magneli Phases. *Chem. Rec.* **18**, 1105-1113 (2018).
- 23 Stoyanov, E., Langenhorst, F. & Steinle-Neumann, G. The effect of valence state and site geometry on Ti L_{3,2} and O K electron energy-loss spectra of Ti_xO_y

- phases. *Am. Mineral.* **92**, 577-586 (2007).
- 24 Holade, Y. *et al.* Electrocatalytic and Electroanalytic Investigation of Carbohydrates Oxidation on Gold-Based Nanocatalysts in Alkaline and Neutral pHs. *J. Electrochem. Soc.* **165**, H425-H436 (2018).
- 25 Men, Y.-L., Liu, P., Liu, Y., Meng, X.-Y. & Pan, Y.-X. Noble-Metal-Free WO₃-Decorated Carbon Nanotubes with Strong W–C Bonds for Boosting an Electrocatalytic Glucose Oxidation Reaction. *Ind. Eng. Chem. Res.* **61**, 4300-4309 (2022).
- 26 Neha, N. *et al.* Revisited Mechanisms for Glucose Electrooxidation at Platinum and Gold Nanoparticles. *Electrocatalysis* 1-10 (2022).
- 27 Bae, I., Yeager, E., Xing, X., Liu, C. J. J. o. e. c. & electrochemistry, i. In situ infrared studies of glucose oxidation on platinum in an alkaline medium. *J. Electroanal. Chem.* **309**, 131-145 (1991).
- 28 Luo, L. *et al.* Selective Photoelectrocatalytic Glycerol Oxidation to Dihydroxyacetone via Enhanced Middle Hydroxyl Adsorption over a Bi₂O₃-Incorporated Catalyst. *J. Am. Chem. Soc.* **144**, 7720-7730 (2022).
- 29 Zhang, X., Chan, K.-Y., You, J.-K., Lin, Z.-G. & Tseung, A. C. J. J. o. E. C. Partial oxidation of glucose by a Pt|WO₃ electrode. **430**, 147-153 (1997).
- 30 Dikalov, S. I., Mason, R. P. J. F. R. B. & Medicine. Reassignment of organic peroxy radical adducts. *J. Electroanal. Chem.* **27**, 864-872 (1999).
- 31 Sato, S. & White, J. M. J. J. o. t. A. C. S. Photoassisted water-gas shift reaction over platinumized titanium dioxide catalysts. *J. Am. Chem. Soc.* **102**, 7206-7210 (1980).
- 32 Rafaideen, T., Baranton, S. & Coutanceau, C. Highly efficient and selective electrooxidation of glucose and xylose in alkaline medium at carbon supported alloyed PdAu nanocatalysts. *Appl. Catal. B* **243**, 641-656 (2019).
- 33 Liang, D. *et al.* Selective oxidation of glycerol with oxygen in a base-free aqueous solution over MWNTs supported Pt catalysts. *Appl. Catal. B* **106**, 423-432 (2011).
- 34 Li, X., Zhang, L., Wang, S. & Wu, Y. Recent Advances in Aqueous-Phase

- Catalytic Conversions of Biomass Platform Chemicals Over Heterogeneous Catalysts. *Front Chem.* **7**, 948 (2019).
- 35 Moggia, G., Kenis, T., Daems, N. & Breugelmans, T. J. C. Electrochemical oxidation of d - glucose in alkaline medium: Impact of oxidation potential and chemical side reactions on the selectivity to d - gluconic and d - glucaric acid. *Electrochim. Acta* **7**, 86-95 (2020).
- 36 Peter, L. M. & Wijayantha, K. G. U. Photoelectrochemical Water Splitting at Semiconductor Electrodes: Fundamental Problems and New Perspectives. *Chemphyschem* **15**, 1983-1995 (2014).
- 37 Gao, Y. & Hamann, T. W. Quantitative hole collection for photoelectrochemical water oxidation with CuWO₄. *Chem. Commun.* **53**, 1285-1288 (2017).
- 38 Cachet, H. & Sutter, E. M. M. Kinetics of Water Oxidation at TiO₂ Nanotube Arrays at Different pH Domains Investigated by Electrochemical and Light-Modulated Impedance Spectroscopy. *J. Phys. Chem. C* **119**, 25548-25558 (2015).
- 39 Peter, L. M. Energetics and kinetics of light-driven oxygen evolution at semiconductor electrodes: the example of hematite. *J. Solid State Electrochem.* **17**, 315-326 (2013).
- 40 Ponomarev, E. A. & Peter, L. M. A Comparison of Intensity-Modulated Photocurrent Spectroscopy and Photoelectrochemical Impedance Spectroscopy in a Study of Photoelectrochemical Hydrogen Evolution at P-Inp. *J. Electroanal. Chem.* **397**, 45-52 (1995).

REVIEWERS' COMMENTS

Reviewer #1 (Remarks to the Author):

Authors should determine that no spontaneous reaction occurs of high concentration KOH with glucose. As far as I know, glucose will undergo a spontaneous reaction in 1M KOH.

The separation of products is critical for the potential application of this system, which I have suggested in the previous round. The authors claim that the K⁺ glucarate salt can be separated through this ion exchange resin method. However, Only the content of K⁺ ions is provided in the ion chromatography, and the real separation problem is not solved.

Is there any hydrogen produced at the anode?

Reviewer #2 (Remarks to the Author):

I am satisfied with the experiments supplemented and changes made in the manuscript, and thus this manuscript could be accepted for publication in Nature Communications. However, my concern remains in the lack of overall evaluation of the manuscript proposed by the three reviewers and the corresponding response, without which appropriate evaluation could not be given.

Reviewer #3 (Remarks to the Author):

The authors have adequately revised the manuscript based on the reviewer's comments. I think that the manuscript in its present form could be accepted for publishing.

Responses to Reviewers

Reviewer 1:

Comment 1: Authors should determine that no spontaneous reaction occurs of high concentration KOH with glucose. As far as I know, glucose will undergo a spontaneous reaction in 1M KOH.

Response: The glucose indeed will undergo a spontaneous reaction in high concentration alkali ($\text{pH} > 11$) solution at high temperature ($T > 50\text{ }^\circ\text{C}$).¹ In our case, the reaction was conducted at $20\text{ }^\circ\text{C}$ and we would expect that the glucose decomposition at this temperature is negligible. To confirm this, the possible glucose decomposition in 1M KOH at $20\text{ }^\circ\text{C}$ was investigated by 2D-HMBC NMR. **Fig. R1a** shows the typical 2D-HMBC NMR of glucose in an aqueous solution.² After 10 h, no obvious changes were observed in the 2D-HMBC NMR spectrum (**Fig. R1b**), suggesting no significant spontaneous reaction of glucose. This is consistent with previous studies of glucose oxidation in alkali solutions.³⁻⁵

Fig. R1 ^{13}C - ^1H HMBC-NMR spectra of glucose solutions (a) in its initial form (10 mM glucose in 1 M KOH) and (b) keeping after 10 h.

Changes to the manuscript: **Fig. R1** and its corresponding description have been added in the Supporting Information as **Supplementary Fig. 25**. The following sentence has been added to the paragraph on Page 18: “The possible glucose decomposition in 1M KOH at 20°C was investigated by 2D-HMBC NMR. No obvious changes were observed in the 2D-HMBC NMR spectra of the solutions before and after keeping for 10 h (Supplementary Fig. 25), suggesting no significant self-decomposition of glucose. This is consistent with previous studies of glucose oxidation in alkali solutions³⁻⁵.”

Comment 2: The separation of products is critical for the potential application of this system, which I have suggested in the previous round. The authors claim that the K+ glucarate salt can be separated through this ion exchange resin method. However, Only the content of K+ ions is provided in the ion chromatography, and the real separation problem is not solved.?

Response: We thank the reviewer's constructive suggestions. The free-glucuronic acid can be separated through ion exchange resin, separation by boronic acid affinity gel, and azeotrope drying. Briefly, after the 5.5 h PEC glucose oxidation from the Pt/def-TiO₂ photoanode, the 20 ml electrolyte was first diluted ten times, and then the obtained solution was mixed with 5 g Amberlyst-15 (H⁺) ion exchange resin.⁶ After 10 mins of ion exchange resin, the collected solution was mixed with boronic acid affinity gel (Affi-Gel boronate gel; Bio-Rad Laboratories, Hercules, CA), and washed with 0.08 M potassium phosphate-0.02 M boric acid buffer (pH 7.0). The GLA was eluted with 0.1 M HCl. The obtained GLA solution was subsequently diluted twenty times with acetonitrile and then the solvent was recovered by rotary evaporation (50 mbar, 22 °C) to yield white powders. The XRD pattern of the collected white powders is consistent with the standard spectra from crystallographic indices derived through single-crystal X-ray diffraction (**Fig R2**),⁶ indicating the purity of the collected GLA.

Fig. R2. XRD patterns of the products collected from the Pt/def-TiO₂ photoanode and the corresponding GLA reference.

Changes to the manuscript: Fig. R2 and its corresponding description have been added in the Supporting Information as **Supplementary Fig. 12**. The following sentence has been added to the paragraph on Page 13: “The glucose conversion (X_{glucose}) from the Pt/def-TiO₂ photoanode was 98.8 % after a 5.5 h reaction, with GLU and GLA yields (Y_{GLU} and Y_{GLA}) of 9.2 and 84.3 %, respectively (Entry 7, **Table 1**), from which free-glucaric acid can be separated through ion exchange resin, separation by boronic acid affinity gel and azeotrope drying (Supplementary Fig. 12).”

Comment 3: Is there any hydrogen produced at the anode?

Response: The gas products from the anode and cathode chambers were separately detected using an online gas chromatograph with Ar as the carrier gas. The results show that no H₂ is produced at the Pt/def-TiO₂ photoanode.

Changes to the manuscript: The following sentences have been added to the paragraph on Page 13: “Besides, the H₂ evolution at both Pt/def-TiO₂ photoanode and counter electrode was examined by online gas chromatography with Ar as carrier gas. The results show that H₂ was undetectable at the Pt/def-TiO₂ photoanode and only detected at the cathode electrode, meaning that the PEC oxidation of glucose on the photoanode could be a water-involved process.³⁻⁵”

Reviewer 2:

Comment: I am satisfied with the experiments supplemented and changes made in the manuscript, and thus this manuscript could be accepted for publication in Nature Communications.

Response: We thank the reviewer for his/her positive comments.

Reviewer 3:

Comment: The authors have adequately revised the manuscript based on the reviewer's comments. I think that the manuscript in its present form could be accepted for publishing.

Response: We thank the reviewer for his/her positive comments.

Reference

- 1 Marianou, A. A. *et al.* Glucose to Fructose Isomerization in Aqueous Media over Homogeneous and Heterogeneous Catalysts. *ChemCatChem* **8**, 1100-1110 (2016).
- 2 Armstrong, R. D., Hirayama, J., Knight, D. W. & Hutchings, G. J. Quantitative Determination of Pt- Catalyzed d-Glucose Oxidation Products Using 2D NMR. *ACS Catal.* **9**, 325-335 (2018).
- 3 Liu, W. J. *et al.* Efficient electrochemical production of glucaric acid and H₂ via glucose electrolysis. *Nat. Commun.* **11**, 265 (2020).
- 4 Aoun, S. B. *et al.* Effect of metal ad-layers on Au (1 1 1) electrodes on electrocatalytic oxidation of glucose in an alkaline solution. *J. Electroanal. Chem.* **567**, 175-183 (2004).
- 5 Pasta, M., Ruffo, R., Falletta, E., Mari, C. & Pina, C. D. J. G. b. Alkaline glucose oxidation on nanostructured gold electrodes. *Gold Bull.* **43**, 57-64 (2010).
- 6 Armstrong, R. D., Kariuki, B. M., Knight, D. W. & Hutchings, G. J. How to Synthesise High Purity, Crystalline d-Glucaric Acid Selectively. *Eur. J. Org. Chem.* **2017**, 6811-6814 (2017).